# The transcription factor Slug represses p16$^{Ink4a}$ and regulates murine muscle stem cell aging

Pei Zhu[1], Chunping Zhang[1], Yongxing Gao[2,3], Furen Wu[1,3], Yalu Zhou[1,3] & Wen-Shu Wu[1]

Activation of the p16$^{Ink4a}$-associated senescence pathway during aging breaks muscle homeostasis and causes degenerative muscle disease by irreversibly dampening satellite cell (SC) self-renewal capacity. Here, we report that the zinc-finger transcription factor *Slug* is highly expressed in quiescent SCs of mice and functions as a direct transcriptional repressor of *p16$^{Ink4a}$*. Loss of *Slug* promotes derepression of p16$^{Ink4a}$ in SCs and accelerates the entry of SCs into a fully senescent state upon damage-induced stress. *p16$^{Ink4a}$* depletion partially rescues defects in *Slug*-deficient SCs. Furthermore, reduced *Slug* expression is accompanied by p16$^{Ink4a}$ accumulation in aged SCs. *Slug* overexpression ameliorates aged muscle regeneration by enhancing SC self-renewal through active repression of *p16$^{Ink4a}$* transcription. Our results identify a cell-autonomous mechanism underlying functional defects of SCs at advanced age. As p16$^{Ink4a}$ dysregulation is the chief cause for regenerative defects of human geriatric SCs, these findings highlight *Slug* as a potential therapeutic target for aging-associated degenerative muscle disease.

[1] Department of Medicine, College of Medicine, University of Illinois at Chicago, Chicago, IL 60612, USA. [2] Department of Medicine, The Johns Hopkins University School of Medicine, Baltimore, MD 21205, USA. [3] These authors contributed equally: Yongxing Gao, Furen Wu, Yalu Zhou. Correspondence and requests for materials should be addressed to W.-S.W. (email: wuwenshu@uic.edu)

Skeletal muscle is a homeostatic tissue that is capable of making up turnover caused by daily wear-and-tear as well as regenerating upon damage. However, such homeostasis declines with aging. Sarcopenia, featured with progressive loss of muscle mass and strength, is one of the most common health issues for individuals at advanced age[1]. Therefore, determining the mechanisms that underlie skeletal muscle aging becomes essential for clinical therapy for the degenerative muscle disease of this kind.

To date, the key factors determining aging-associated decline of muscle regenerative capacity has remained open to debate. It is believed that proper muscle regeneration necessitates optimal extrinsic environmental supplies. Direct evidence supporting this viewpoint came from a whole muscle grafting experiment demonstrating that the regenerative ability of young and old muscles depended mainly on the age of the host[2]. More recent findings of increasing fibroblast growth factor (FGF) signaling in aged muscle fiber[3], as well as the circulating protein growth differentiation factor 11 as a rejuvenating factor for aged skeletal muscle[4] further highlighted the importance of local and systemic environment in muscle aging, respectively. In contrast, muscle stem cells (MuSCs) purified from aged mice show a substantial decline in the number of MuSCs that engraft and regenerate in recipient muscle[5–7], shedding light on the role of stem cell intrinsic changes in regulation of muscle aging. Indeed, restoring the disordered signaling pathway, such as p38 MAPK[6], in aged MuSCs rejuvenate MuSC function and muscle regeneration in old mice.

Regardless of extrinsic environment or cell-autonomous mechanisms, aging-associated changes lead to stem cell exhaustion in the elderly. Pax7-expressing satellite cells (SCs) function as a necessary stem cell population responsible for growth, maintenance, and regeneration of skeletal muscle[8,9]. Cellular senescence is an important cause of stem cell exhaustion with aging in multiple tissues. Telomere shortening, nontelomeric DNA damage and derepression of the $INK4/ARF$ locus are all able to induce senescence[10–12]. Notably, derepression of $p16^{Ink4a}$ switches geriatric SCs from reversible quiescence into senescence, leading to incompetency of activation on muscle injury even in a youthful environment[5]. However, those cell intrinsic components regulating $p16^{Ink4a}$ expression in SCs remain largely unknown.

In this study, we establish $Slug$, a member of zinc-finger transcription factor in the Slug/Snail superfamily, as a transcriptional repressor of $p16^{Ink4a}$ in SCs. Akin to mice with aging, loss of $Slug$ endows adult SCs features of pre-senescence by largely inducing $p16^{Ink4a}$ expression, triggering apparent regenerative defects during serial muscle damage. Importantly, reduced $Slug$ and elevated $p16^{Ink4a}$ expressions in SCs simultaneously occur with chronological aging. Restoration of $Slug$ expression is capable of rejuvenating aged SC functions. Our results highlight $Slug$ as a key target for aging-associated degenerative muscle disease.

## Results

**$Slug$ deletion causes a defect in muscle regeneration.** The transcription factor $Slug$ is expressed in a variety of normal tissues in the adult mouse[13], indicating its important roles in development. In agreement with this notion, $Slug$ knockout mice showed numerous abnormalities such as smaller body size and weight (Supplementary Fig. 1a,b). Since skeletal muscle accounts for ~40% of adult body weight[9], we examined if the reduced body weight in $Slug$-deficient mice is due to a reduction of muscle mass. Hindlimb muscles of adult $Slug$-mutant mice were reduced by 21–36% in weight (Supplementary Fig. 1c). To some extent, the lost muscle mass was caused by reduction in myofiber size in

$Slug$-deficient mice (Supplementary Fig. 1d, e and Fig. 1e). However, when normalized to body weight, none of the relative weights of these muscles were affected by the absence of $Slug$ (Supplementary Fig. 1f). Changes in muscle mass may be resulted from changes in protein or cell turnover[14]. The latter reflects the balance between myonuclear accretion and loss. Proliferation and fusion of SCs increases the number of myonuclei within the muscle fibers. Therefore, we determined the effect of $Slug$ deficiency on MuSC maintenance. Unexpectedly, $Slug$-deficient mice had even slightly higher fraction of SCs in non-lineage cell subpopulation (Fig. 1a). The total yield of SCs calculated as per milligram of harvested hindlimb muscle was also increased in $Slug$ knockout mice (Fig. 1b). Such $Slug$ ablation-induced increases in MuSC frequency and number were further confirmed by staining of Pax7$^+$ nuclei on freshly prepared TA muscle cryosections (Fig. 1c, d).

SCs is essential for the maintenance and regeneration of skeletal muscle. Thus, we determined how $Slug$ null-induced increase of SCs affects muscle regeneration upon injury. H&E staining showed that $Slug$-deficient muscles regenerated as well as that of wild-types after a single injury (Fig. 1e–g). Strikingly, upon second injury, $Slug$-deficient muscles exhibited severely impaired regeneration when compared with wild-type counterparts (Fig. 1e–g). Collectively, these data demonstrated that Slug is essential for efficient muscle repair during continuous muscle regeneration, suggesting a key role of $Slug$ in regulation of SC function.

**SC-specific $Slug$ loss impairs skeletal muscle regeneration.** Skeletal muscle regeneration is a highly coordinated process involving the activation of various cellular and molecular responses[15]. We first decided to examine the expression pattern of $Slug$ in SCs in view of their critical role in muscle regeneration. By comparing with several other muscle resident cell types including fibro-adipogenic progenitors (FAPs), pan-lymphocytes (LCs), and epithelial cells (ECs), in which the role of $Slug$ is well characterized, we found that $Slug$ is most highly expressed in quiescent SCs (Fig. 2a), and its expression was slightly reduced in activated SCs and markedly decreased after SCs were differentiated into myotubes (Fig. 2b).

To determine whether the impaired muscle regeneration in global $Slug$ knockout mice is a SC-driven defect, we generated SC-specific $Slug$ knockout mouse line using the Cre/loxP system (Fig. 2a–d and Supplementary Fig. 2). Unlike the global $Slug$ knockout mice, $Slug^{fl/fl}Pax7^{Cre/+}$ (designated as $Slug^{cKO}$) mice showed no apparent differences with the control animals ($Slug^{fl/+}/Pax7^{Cre/+}$) in body size and weight (Fig. 2e). Furthermore, SCs from Ctrl but not $Slug^{cKO}$ mice displayed positive-staining for Slug protein, indicating that $Slug$ is efficiently deleted in SCs in $Slug^{cKO}$ mice (Fig. 2f).

Next, we investigated the effect of SC-specific $Slug$ deletion on the maintenance and regenerative capacity of SCs. Consistently, both the overall frequency and total yield of SCs calculated as per milligram of muscles were increased in $Slug^{cKO}$ mice (Fig. 2g, h). Although individual myofiber diameters of the intact TA and cross-sectional area of TA on day 10 post single BaCl$_2$ injury were not different between Ctrl and mutant littermates, more severely compromised muscle regeneration with a large increase of both necrotic fibers and fibrotic tissue was observed in $Slug^{cKO}$ mice administered with double and triple muscle damages (Fig. 2i, j). The impaired secondary but not primary muscle regeneration was also detected in $Slug^{fl/fl}Pax7^{CreER}$ mice, in which $Slug$ was deleted specifically in adult SCs by administering tamoxifen prior to muscle injury (Supplementary Fig. 3). Taken together, these results demonstrated that the presence of $Slug$ in SCs is essential

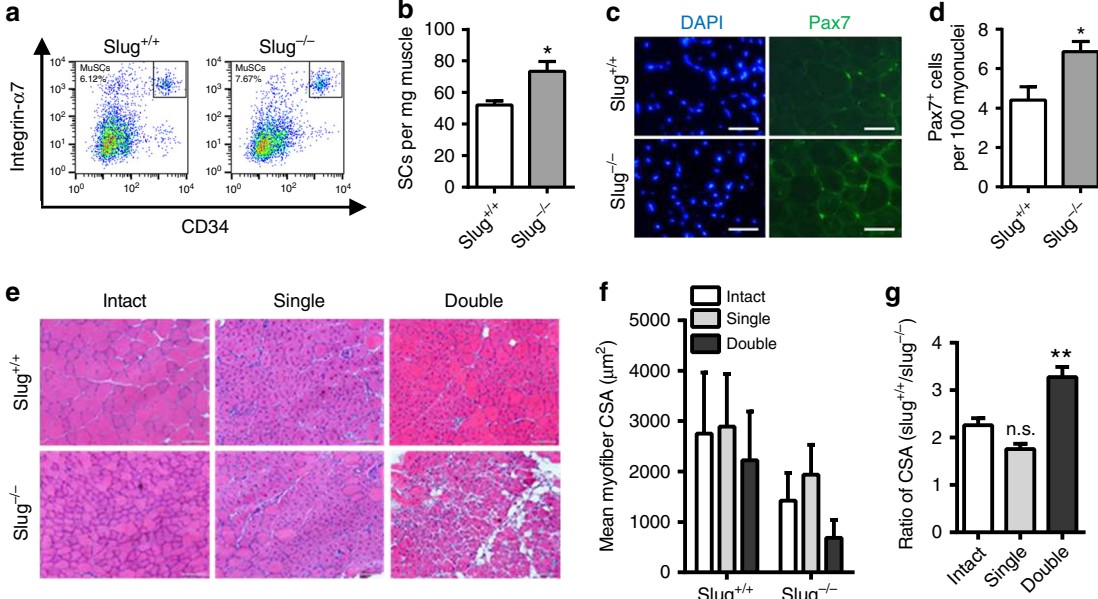

**Fig. 1** *Slug* deficiency repairs regenerative capacity of SCs during serial muscle damage. **a** Representative flow cytometric analysis of the frequency of SCs ($CD45^-$/$CD11b^-$/$CD31^-$/$Sca1^-$/Integrin-$\alpha7^+$/$CD34^+$) subpopulation in $Slug^{+/+}$ and $Slug^{-/-}$ mice. **b** Yield of SCs per milligram (mg) of muscle from $Slug^{+/+}$ and $Slug^{-/-}$ mice ($n = 6$ mice per group). $^*p < 0.05$ by student's $t$-test. **c** Representative IHC images of $Pax7^+$ SCs within the intact tibialis anterior (TA) muscles of $Slug^{+/+}$ and $Slug^{-/-}$ mice. DAPI was used as nuclear counterstaining. Scale bar, 100 μm. **d** Quantification of $Pax7^+$ SC numbers in **c**. $^*p < 0.05$ by student's $t$-test. **e** H&E staining of the intact and injured TA muscles in $Slug^{+/+}$ and $Slug^{-/-}$ mice ($n = 5$ mice per group). For single injury, TA muscles were harvested at day 10 after $BaCl_2$ injection. For double injury, mice were recovered for 1 month after the first $BaCl_2$ injection at TA muscles. A second $BaCl_2$ injection was administered thereafter. TA muscles were harvested 10 days after the second $BaCl_2$ injection. Scale bar, 100 μm. **f** Quantification of the mean myofiber cross-sectional area (CSA, μm²) of the intact and $BaCl_2$-injured TA muscles of $Slug^{+/+}$ and $Slug^{-/-}$ mice. **g** Ratio of myofiber CSA in the intact and $BaCl_2$-injured TA muscles between $Slug^{+/+}$ and $Slug^{-/-}$ mice. $^{**}p < 0.01$ by student's $t$-test (n.s., not significant). All these experiments were independently repeated three times with similar results. Data are presented as mean ± SEM. Also see Supplementary Fig. 1. Source data are provided as a Source Data file

for SC-driven skeletal muscle regeneration, and *Slug* regulates muscle regeneration via its SC-specific function.

**Slug-null SCs fail to replenish SC pool after activation.** Maintaining SC pool size is crucial for constant muscle turnover/injuries and ongoing repair[16]. We showed that muscle regeneration after first injury was normal, indicating that *Slug*-deleted SCs in intact muscles were functional in terms of activation and differentiation upon injury. Indeed, primary $Slug^{-/-}$ SCs showed robust potential for differentiation upon *in vitro* induction and *in vivo* transplantation (Supplementary Fig. 4). However, the consequent regeneration became severely compromised in the absence of *Slug* (Figs. 1e, 2i and Supplementary Fig. 3e). These results prompted the question whether *Slug*-deficient SCs were capable of self-renewing and replenishing the stem cell pool after activation. $Slug^{cKO}$ displayed a sharp decrease in SC number of regenerated TA muscle when compared with Ctrl littermates after single and double injuries (Fig. 3a), indicating that lack of *Slug* attenuates ability of SCs to properly maintain the stem cell pool in repaired muscle after injury. This was confirmed by an observation of about three-fold decrease in SC number per 100 myonuclei in regenerated TAs of $Slug^{cKO}$ mice when compared with control mice (Fig. 3b, c).

Injection of tractable adult SCs into pre-injured adult muscle followed with fluorescence-activated cell sorting (FACS) analysis provides a quantitative assay for SC self-renewal[17]. By this assay, we analyzed the stem cell repopulation of donor-derived SCs ($GFP^+$) as a fraction of the total SC population from the primary recipients (Fig. 3d). Because a small portion of SCs remain cycling 1 month after transplantation[18], we analyzed SCs with distinct immunophenotype ($CD31^-CD45^-Sca1^-Vcam\ I^+$), which consist of both

quiescent and activated MuSCs[19]. Via this analysis, we found that the frequency of $Slug^{-/-}$ SCs was about three-fold lower than that of $Slug^{+/+}$ SCs (Fig. 3e, f). Since the compromised self-renewing capability of SCs from $Slug^{-/-}GFP$ mice may be due to embryonic absence of *Slug*, we induced an acute *Slug* knockout in SCs from $Slug^{fl/fl}Pax7$-zsgreen mice by infecting with retrovirus expressing Cre recombinase (Supplementary Fig. 5a,b). Cre-expressing SCs yielded about five-fold less $GFP^+$ fraction in the total SC population from the recipients compared to that of Ctrl virus-infected SCs (Supplementary Fig. 5c,d). Consistently, by zsgreen staining we identified considerably less Cre retrovirus-infected SCs in the SC niche, beneath the basal lamina and atop myofibers ($p < 0.01$ by student's $t$-test) (Supplementary Fig. 5e,f). These data indicated that deletion of *Slug* decreases SC self-renewing and regenerative ability.

Competitive repopulation assay has been widely used as the gold standard for testing self-renewal capacity of hematopoietic stem cells[20]. Here, we adopted a similar competitive repopulation assay for SCs (Fig. 3g). Technically, this assay overcomes the potential variations due to different SC niches in individual recipients and uneven injection of myotoxins and SCs at different sites. By this assay, we showed that although donor SCs from $Slug^{+/+}GFP$ and $Slug^{-/-}GFP$ mice were equally mixed before transplantation, only the DNA band corresponding to wild-type alleles of *Slug* was predominantly amplified from the gDNA of the donor-derived SC mixtures in recipients (Fig. 3h and Supplementary Fig. 6), indicating a markedly attenuated self-renewing capability in $Slug^{-/-}$ SCs.

To further assess the intrinsic self-renewing propensity of *Slug* null SCs, we performed myofiber-associated SCs culture in the presence of arabinosylcytosine (AraC) that eliminates cycling cells by inhibiting DNA synthesis. It was previously demonstrated

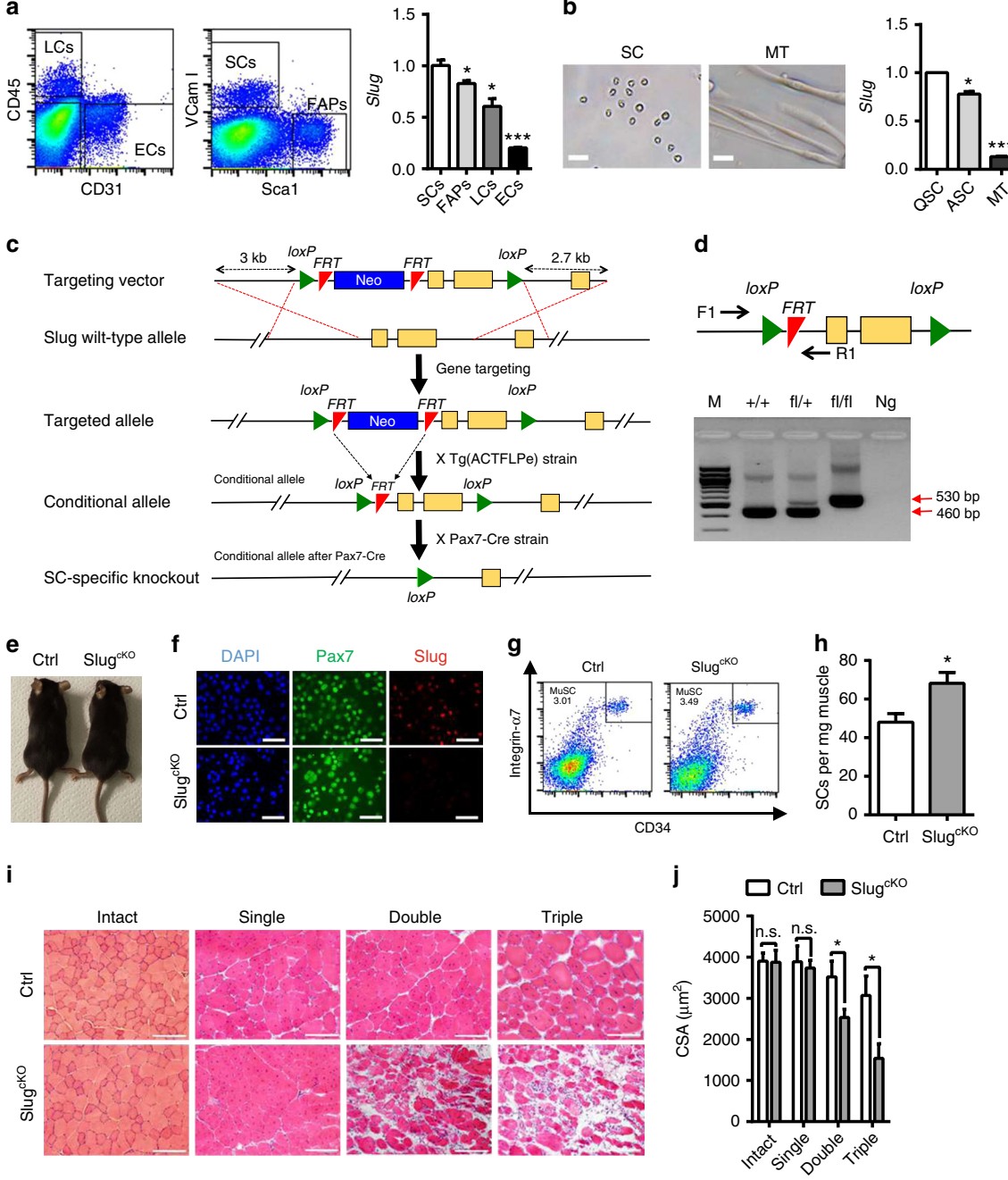

**Fig. 2** SC-specific Loss of *Slug* Impairs Muscle Regeneration. **a** *Slug* expression in different muscle resident cells. Left, representative flow cytometric gating of SCs (CD31⁻CD45⁻Scal1⁻Vcam I⁺), pan-lymphocytes (LCs, CD45⁺), epithelial cells (ECs, CD31⁺), and fibro-adipogenic progenitors (FAPs, CD31⁻CD45⁻Scal1⁺) from freshly prepared skeletal muscle cells. Right, qPCR analysis of *Slug* expression. *$p < 0.05$, **$p < 0.01$, ***$p < 0.001$ by student's *t*-test. **b** Quantification of *Slug* expression in undifferentiated and differentiated SCs. Left, representative images of SCs and myotubes. Scale bar, 100 μm. Right, qPCR analysis of *Slug* expression. *$p < 0.05$, ***$p < 0.001$ by student's *t*-test. QSC, quiescent satellite cell; ASC, activated satellite cells upon culture in growth medium for 3 days; MT, myotube. **c** Gene targeting strategy for generation of SC-specific *Slug* knockout mice. **d** Diagram of *Slug*-specific primer design for genotyping PCR. M. DNA marker; Ng negative control for PCR. **e** Comparison of adult *Slugfl/flPax7Cre/+* (*SlugcKO*) and *Slugfl/+Pax7Cre/+* (Ctrl). **f** Immunofluorescence staining of Slug in SCs of *SlugcKO* and Ctrl mice (n = 3 mice). Scale bar, 100 μm. **g** Frequency of SCs in *SlugcKO* and Ctrl mice. Similar results were seen from three independent flow cytometric analyses. **h** Yield of SCs per mg of muscle from Ctrl and *SlugcKO* mice (n = 3 mice for each genotype). *$p < 0.05$ by student's *t*-test. **i** H&E staining of intact and injured TA muscles in *SlugcKO* and Ctrl mice (n = 5 mice per group). For single injury, TA muscles were harvested at day 10 after BaCl₂ injection. For consecutive injury, mice with primary injury were recovered for 1 month followed by a second BaCl₂ injection at the same sites. TA muscles were harvested 10 days after each injury. Scale bar, 100 μm. The experiment was repeated independently for three times with similar results. **j** Quantification of the myofiber CSA (μm²) shown in **i**. *$p < 0.05$ by student's *t*-test (n.s., not significant). Data are shown as mean ± SEM of three independent replicates. Also see Supplementary Fig. 2 and 3. Source data are provided as a Source Data file

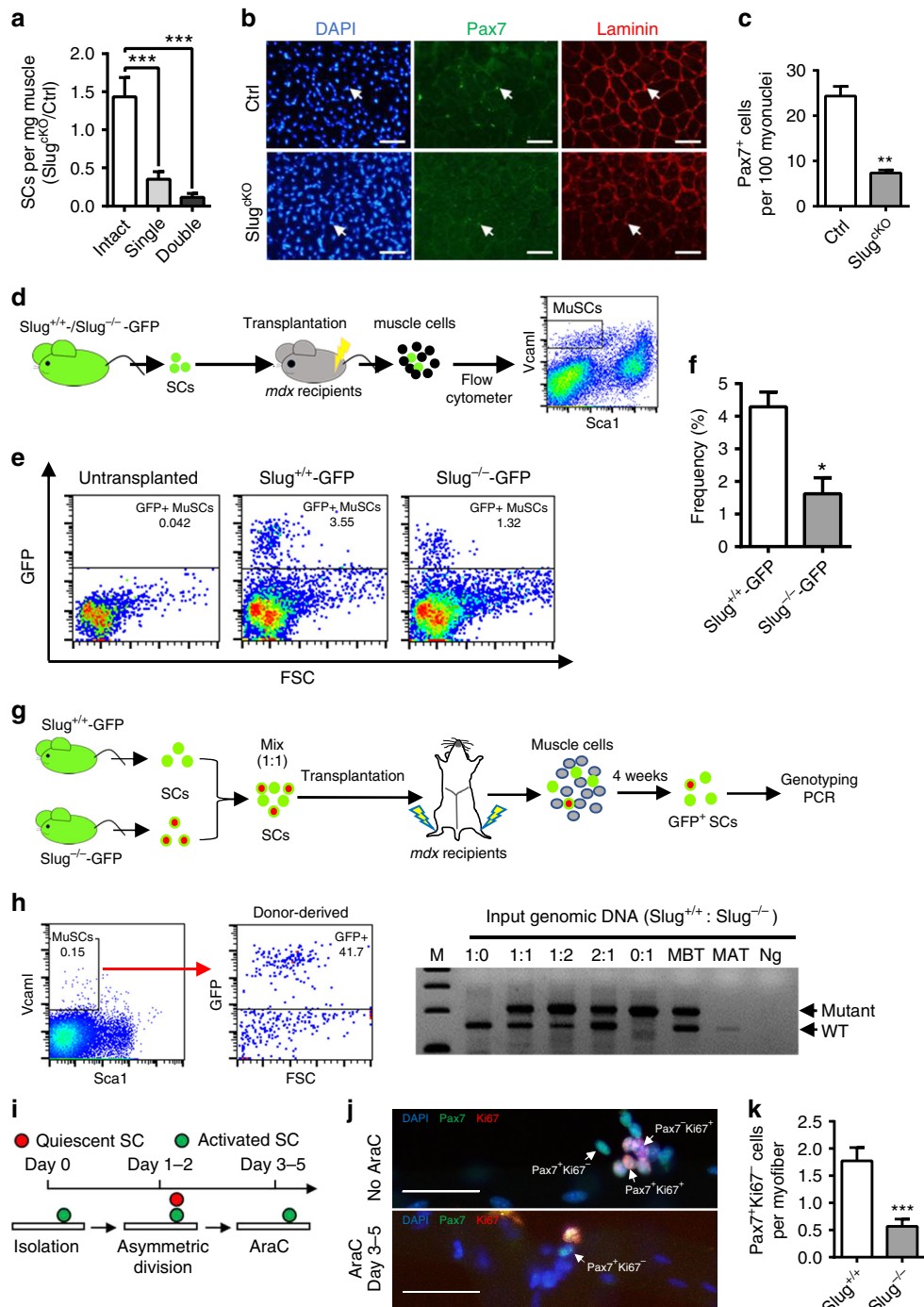

that a small population of AraC-resistant, myofiber-associated Pax7[+] cells arose following the first SC division in culture by self-renewing and behaved as quiescent SCs[6,21]. We treated myofibers with AraC from day 3–5 after isolation (Fig. 3i) and detected a lower number of surviving Pax7[+]Ki67[−] SC daughters from *Slug^cKO* mice compared to those from Ctrl mice (Fig. 3j, k). Together, these results provide convincing evidence that *Slug*-deficient SCs have an intrinsic defect in self-renewal following muscle regeneration.

**Slug directly represses *p16^Ink4a* transcription in SCs**. To explore intrinsic factors responsible for losing self-renewal capacity of activated SCs in *Slug*-deficient mice, we initially retrieved and interrogated the global gene expression data (GEO accession:

GSM38236) generated from *Slug*-silenced primary myoblasts[22]. Gene ontology enrichment analysis of biological processes (GOBP) identified that *Slug* silencing derepressed sets of genes related to transcription regulation, cell proliferation, and skeletal muscle differentiation processes (Fig. 4a, left panel). Notably, the genes downregulated upon *Slug* silencing identified enrichment of genes among various categories of cellular defense responses (Fig. 4a, right panel). Interestingly, p16^Ink4a was listed in the most upregulated genes ranked in negative regulation of cell proliferation. It was previously reported that self-renewal capacity of aged stem cells (hematopoietic stem cells[23], intestinal stem cells[24], and skeletal MuSCs[5]) was attenuated. Therefore, we postulated p16^Ink4a as a potential mediator for SC self-renewing defect caused by *Slug* deficiency.

**Fig. 3** SCs deficient in *Slug* exhibit intrinsic defects in self-renewal in vivo. **a** Ratio of SCs between *Slug^cKO* and Ctrl mice under indicated conditions. ***p < 0.001 by student's *t*-test. **b** IHC for Pax7+ SCs in TAs of *Slug^cKO* and Ctrl mice (n = 3 mice per group) at day 30 post single injury. Laminin indicates the boundaries of myofibers. Scale bar, 100 μm. Three independent experiments were performed with similar results. **c** Quantification of Pax7+ SC numbers in **b** **p < 0.01 by student's *t*-test. **d** Scheme of SC transplantation. 3000 SCs of indicated genotype were transplanted into each side of pre-injured TA muscles of *mdx* recipients (n = 6), respectively. Donor-derived SCs (GFP+) in total recipient MuSCs (CD45−CD31−Sca1−Vcam I+) were analyzed 4 weeks after transplantation. **e** Representative flow cytometric analysis of the frequency of donor-derived SCs (GFP+) within total recipient MuSCs. Similar results were seen in three independent transplantation experiments. **f** Percent of donor-derived SCs (GFP+) in total recipient MuSCs. *p < 0.05 by student's *t*-test. **g** Diagram of competitive MuSC repopulation assay. SCs from *Slug^+/+GFP-* and *Slug^−/−GFP*-transgenic mice were equally mixed for genomic DNA extraction and transplantation, respectively. The remaining cells were then transplanted into pre-injured TA muscles of *mdx* recipients. Four weeks after transplantation, GFP+ cells were sorted from total recipient MuSCs for genotyping PCR analysis. **h** Determination of relative ratio of repopulated MuSCs. Left, representative flow cytometric plotting of GFP+ cells from total recipient MuSCs. Right, PCR determining the ratio of SCs before and after transplantation. M DNA ladder, MBT mixed SCs before transplantation, MAT mixed SCs after transplantation, Ng negative control. **i** Scheme of AraC treatment indicating self-renewal. **j** Pax7 and Ki67 immunostaining in cultured myofibers as treated in **i**. Scale bar, 50 μm. Similar results were seen from three independent experiments. **k** Quantification of the quiescent daughter SCs stained in **j** (n = 3 mice, >20 myofibers per condition). ***p < 0.001 by student's *t*-test. Data are shown as mean ± SEM of three independent experiments. Unprocessed gel blots are provided in the Supplementary Fig. 1. Also see Supplementary Fig. 4-6. Source data are provided as a Source Data file

We then performed microarrays analysis to compare the genome-wide gene expression profiles of wild-type and *Slug*-deficient SCs. Unsupervised hierarchical clustering analysis separated the samples into their respective genotypes (Fig. 4b). We identified 168 differentially expressed genes, of which 108 genes were upregulated (fold change > 2, *p* value < 0.05) and 60 genes were downregulated (fold change < −2, *p* value < 0.05) in *Slug^−/−* SC compared to control cells (Fig. 4b). *Slug* deletion enriched GO categories related to mitochondria metabolism and cell cycle regulators (Fig. 4c). To identify pathways enriched in *Slug^−/−* SCs, gene set enrichment analysis (GSEA)[25] was performed. Notably, signaling pathways involved in cell metabolism including glycolysis, PI3K-AKT-MTOR, oxidative phosphorylation, and reactive oxygen species were induced in quiescent *Slug^−/−* SCs (Supplementary Fig. 7a,b), suggesting a switched metabolic reprogramming with relatively higher energy-consuming status in *Slug* null SCs. Meanwhile, E2F targets and G2M checkpoints signaling signatures were also enriched upon *Slug* deletion (Supplementary Fig. 7c,d). These gene sets enrichment data indicated that loss of *Slug* in SCs might disturb cell cycle progression and the balance between self-renewal and differentiation after activation.

Next, we examined expression of *p16^Ink4a* and *p19^Arf*, the two gene products of the INK4A/ARF locus, by qPCR in SCs of adult wild-type and *Slug^−/−* mice, respectively. As shown in Fig. 4d, *p16^Ink4a* was about two-fold higher in *Slug*-deficient SCs when compared with the wild-type counterparts, indicating a derepression of *p16^Ink4a* in the absence of *Slug* in resting SCs in vivo. In contrast, there was no apparent difference in *p19^Arf* expression between the two types of SCs. Furthermore, we examined the proximal promoter region of *p16^Ink4a* and identified potential Slug-binding sites (E-box)[26,27] (Fig. 4e). To facilitate assessing the occupancy of endogenous Slug at the promoter region of *p16^Ink4a* by chromatin immunoprecipitation (ChIP) assay, we performed *Slug* affinity tagging at its C-terminus in mouse myoblasts by CRISPR/Cas9-mediated gene tagging (Supplementary Fig. 8). ChIP-qPCR analysis displayed enriched binding in anti-Flag antibody immunoprecipitated gDNA fragments but not in anti-IgG control (Fig. 4f), indicating that Slug occupies the promoter region of *p16^Ink4a* in vivo. Such direct regulation of *p16^Ink4a* promoter by Slug was further confirmed by *p16^Ink4a*-driven luciferase reporter assay in SC-derived myobalsts (Supplementary Fig. 9a,b). Importantly, this E-box element is also present in human *p16^INK4A* promoter (Supplementary Fig. 9c). Knockout or overexpression of *SLUG* in primary human myoblasts significantly derepressed or suppressed *p16^INK4A* transcript (*p* < 0.001 by student's *t*-test) (Supplementary Fig. 9d-g),

respectively, indicating a highly conserved role of Slug in regulating *p16^Ink4a*.

Of note, loss of *Slug* caused a more robust dysregulation of *p16^Ink4a* in ex vivo cultured myoblasts[22] compared to resting SCs in vivo. Normally, *p16^Ink4a* expression is elevated during aging and replicative senescence[28]. Myoblasts are a type of proliferating myogenic progenitor cells derived from quiescent SCs under stimulating conditions. Therefore, we asked whether derepression of *p16^Ink4a* in *Slug*-deficient SCs would be exacerbated under stress conditions, including ex vivo culture and muscle damage. Strikingly, *p16^Ink4a* expression increased by over 8-fold in ex vivo cultured *Slug^−/−* SCs and five-fold in in vivo injury-activated *Slug^−/−* SCs, respectively, when compared with their corresponding wild-type controls (Fig. 4g) Notably, derepression of *p16^Ink4a* in cultured myoblasts was accompanied with graduate decline of *Slug* expression (Supplementary Fig. 10).

In spite of the increased *p16^Ink4a* transcription, primary muscle regeneration was normal in *Slug*-deficient mice. This is a departure from what was reported in geriatric mice model, i.e., that resting SCs fail to activate and expand to regenerate the muscle on injury due to derepression of *p16^Ink4a*[5]. We suspected that p16^Ink4a mRNA but not protein was increased in resting *Slug*-deficient SCs, since *p16^Ink4a* mRNA could be induced to decay by an RNA-binding protein in early passage of fibroblasts but accumulated at protein level during late-passage of culture[28]. Indeed, p16^Ink4a protein was undetectable in undamaged muscle cryosections from both Ctrl and *Slug^cKO* mice (Supplementary Fig. 11). Instead, there was an apparent co-expression of p16^Ink4a with Pax7 in muscle tissue harvested from *Slug^cKO* mice 30-day post injury (Fig. 4h, i). Taken as whole, these results suggested that *Slug* deficiency leads to an increase in *p16^Ink4a* transcription in SCs, and replicative stress signaling triggered by SC activation and proliferation concurrently increases the stability of p16^Ink4a protein.

**Slug loss promotes senescence in SCs during proliferation.** In terms of the elevated p16^Ink4a protein in activated SCs in regenerating skeletal muscles of *Slug*-deficient mice, we hypothesized that *Slug*-ablated SCs acquired features of cellular senescence at late stages of regeneration during the transition of SCs from activation to quiescence or differentiation. To test this notion, we first used an in vitro reserve cell culture system[29]. As shown in Fig. 5a, reserve cells from wild-type mice robustly proliferated while cells from *Slug^−/−* mice were only sporadically distributed by day 7 of subculture (Fig. 5a). Compared to their wild-type control cells, significantly higher proportion of progeny from

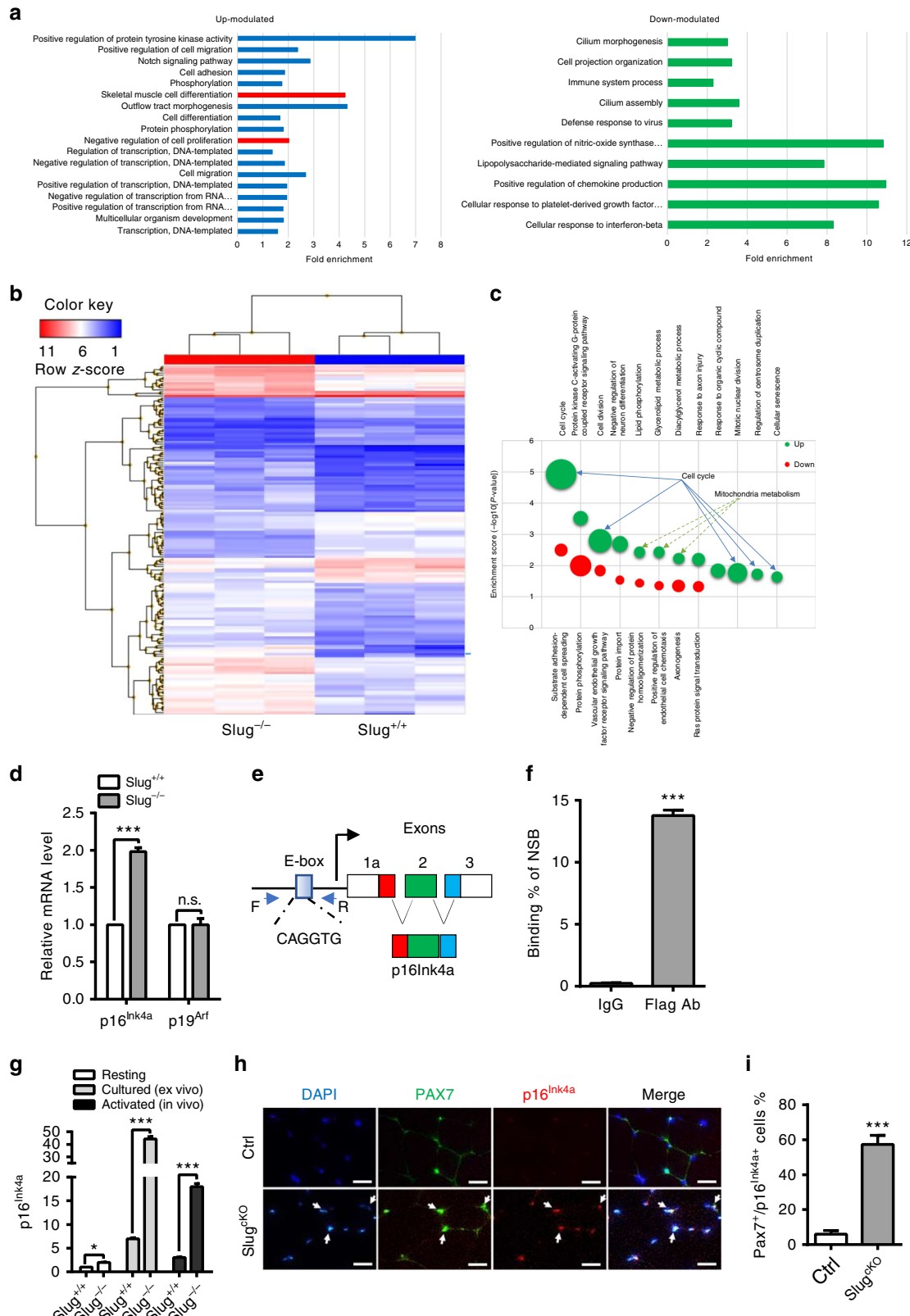

$Slug^{-/-}$ reserve cells was positive for p16$^{Ink4a}$ staining ($p < 0.001$ by student's $t$-test) (Supplementary Fig. 12). SA-β-Gal staining showed that over 50% of $Slug$-deficient reserve cells-expanded cells was SA-β-Gal$^+$, a marker of senescence, whereas few SA-β-Gal$^+$ cells were detected in the control group (Fig. 5a, b).

Population doubling level (PDL) is an intrinsic measurement of the age of the particular culture of a cell line. In culture, an untransformed cell line has a finite life span expressed in the number of cumulative population doublings that can be achieved. To assess the accelerated senescence in $Slug$-deficient SCs under

**Fig. 4** *p16^Ink4a* is a direct target gene of Slug in SCs. **a** GOBP analysis of genes modulated in *Slug*-silenced myoblasts (GEO accession: GSE38236). The term of negative regulation of cell proliferation being highly relevant to observed self-renewal defect of *Slug*-deficient SCs in our study was highlighted in red. **b** Hierarchical clustering and heatmap representation of 168 differentially expressed genes separating *Slug*^+/+ and *Slug*^−/− SCs. Color pattern represents row Z-score. **c** Bubble chart showing results of GOBP analysis. Bubble size indicated number of genes associated with each term. GOBP ranked by fold enrichment score associated with upmodulated (green bubbles) and downregulated (red bubbles) signatures were selected for significance by using a false discovery rate cutoff of 5%. **d** qPCR analysis of *p16^Ink4a* and *p19^Arf* in primary SCs from *Slug*^+/+ and *Slug*^−/− mice (n = 3). n.s. not significant; ***$p < 0.001$ by student's *t*-test. **e** Schematic diagram of the *INK4a/ARF*. The consensus Slug-binding site (E-box) is located in its promoter. **f** ChIP analysis of Slug occupancy at the *p16^Ink4a* promotor. Primers targeting the 3′ untranslated region (without E-box elements) was used as a negative control for the ChIP assay. Relative binding affinity of Slug on the putative E-box element was quantified relative to the non-specific binding. ***$p < 0.001$ by student's *t*-test. **g** Quantification of *p16^Ink4a* in quiescent, *ex-vivo* culture- and *in vivo* injury-activated SCs from *Slug*^+/+ and *Slug*^−/− mice. For activation *ex-vivo*, SCs were cultured for 7 days; while for activation in vivo, SCs were harvested at day 10 after injury. *$p < 0.05$, ***$p < 0.001$ by student's *t*-test. **h** IHC for p16^Ink4a and Pax7 protein in SCs within the TA muscles of *Slug*^cKO and Ctrl mice at day 30 post injury (n = 3 mice per genotype). Scale bar, 100 μm. **i** Percentage of Pax7 and p16^Ink4a double positive SCs in **h**. ***$p < 0.001$ by student's *t*-test. The experiments **g**–**i** were independently repeated three times with similar results. Data are shown as mean ± SEM of three independent experiments. Also see Supplementary Fig. 7-11. Source data are provided as a Source Data file

proliferative pressure in vitro, we determined the relative growth rates of control and *Slug*-mutant SCs by calculating the cumulative PDL. As shown in Fig. 5c, the growth rate of *Slug*-deficient SCs was clearly retarded at early passage 3 when wild-type SCs remained for exponential growth. SA-β-Gal staining demonstrated that about 40% of *Slug*-ablated myoblasts show strong positive SA-β-Gal staining while less than 5% of control cells were SA-β-Gal^+ on passage 3 (Fig. 5d, e), indicating the existence of replicative senescence and growth retardation in early-passaged *Slug*-deficient myoblasts.

Next, we investigated whether cellular senescence in SCs would occur when *Slug* is specifically deleted in Pax7^+ SCs during regeneration in vivo. Young adult mice were used in this study to exclude the process of regeneration from aging in geriatric mice[5]. We performed SA-β-Gal staining on cryosections of TA muscles from Ctrl and *Slug*^cKO mice on day 10 post first injury when necrotic fibers were replaced by central-nucleated regenerated myofibers (Fig. 5f). Notably, despite the sporadic SA-β-Gal^+ cells in control TA muscle sections, there was a four-fold increase in the number of SA-β-Gal^+ cells in *Slug*^cKO TA (Fig. 5f, g). Most of the SA-β-Gal^+ cells were located beneath basal lamina and outside myofiber plasma membrane, a classical SC anatomical location (Fig. 5f). IHC results further demonstrated that these SA-β-Gal^+ cells were also Pax7-positive. As expected, SA-β-Gal^+Pax7^+ cells were not in cycling (Ki67^−), suggesting the status of cellular senescence. In contrast, SCs with negative SA-β-Gal staining (SA-β-Gal^−/Pax7^+) were on the occasion of proliferating (Ki67^+). In agreement with this notion, the majority of SCs in *Slug*^cKO mice failed to re-activate as indicated by markedly lowered percentage of Ki67^+ SCs by 2.5-day post second BaCl₂ injury compared to Ctrl mice (Fig. 5i, j). Together, using multiple in vitro and in vivo assays we proved that lack of *Slug* facilitated entry of SCs into cellular senescence under proliferative pressure.

**p16^Ink4a loss restores impaired self-renewal of *Slug*^−/− SCs.** We demonstrated elevated p16^Ink4a protein and acquired senescence feature in activated *Slug*-deficient SCs. At this point, a key question was whether p16^Ink4a is causally involved in *Slug* loss-induced defects in SC self-renewing capacity and muscle regeneration. To answer this question, we assessed muscle regeneration in *Slug*^+/+ *p16*^+/+, *Slug*^−/−*p16*^+/+, and *Slug*^−/−*p16*^−/− mice following serial damages. By H&E staining, we showed that although muscle regeneration was severely compromised in TAs of *Slug*^−/−*p16*^+/+ mice following double or triple injuries, muscle repair was greatly improved in *Slug*^−/−*p16*^−/− mice after injuries (Fig. 6a, b).

Next, we assessed how removal of p16^Ink4a would affect the self-renewing ability of *Slug*-ablated SCs by the quantitative repopulating assay (Fig. 6c). As expected, the frequency of *Slug*^−/−*p16*^+/+ donor-derived SCs was low (~1.96%) (Fig. 6d, e). However, the frequency of *Slug*^−/−*p16*^−/− donor-derived SCs was increased by two-fold in the recipients (Fig. 6d, e). Mechanistically, enhanced repopulation of *Slug*^−/−*p16*^−/− SC in recipient muscles after deletion of *p16^Ink4a* might be partially ascribed to reduced senescence under proliferative stress, as *Slug*^−/−*p16*^−/− SC-derived myoblasts displayed little or no SA-β-Gal staining at ex vivo passage 3 when majority of *Slug*^−/−- cells were SA-β-Gal positive (Fig. 6f, g). In summary, p16^Ink4a is a crucial mediator for the defects in SC self-renewal and muscle regeneration caused by *Slug* deficiency.

**Slug overexpression restores self-renewal of aged SCs.** Aging alters stem cell function in terms of the capacities of self-renewal, proper activation and/or proliferation as well as lineage commitment. Aged SCs were characterized by self-renewal defect[6,7,16], inclined commitment of differentiation in low mitogen ex vivo culture condition[3,6], and susceptibility to senescence upon mitogen exposure[7,30]. These aging-associated phenotypes resemble to some extent behaviors of *Slug*-deficient SCs in current study. Such similarities drew our attention to the role of *Slug* in SC aging. By interrogating the transcriptomes of young and aged SCs from different groups of mice[4,31], we observed an age-associated downregulation of *Slug* in aged SCs (Fig. 7a). This is further confirmed in SCs from young and aged mice by PCR analysis (Fig. 7b–d).

We demonstrated that there was an increased bioenergetic requirement in *Slug* null SCs (Supplementary Fig. 7a). Interestingly, GSEA revealed a similarly altered metabolic reprogramming in old SCs (GSE81096), which was evidenced by activation of genes ranked in glycolysis and mitochondrial metabolism in MTORC1, oxidative phosphorylation, PI3K-Akt-MTOR, and reactive oxygen species signaling pathways (Supplementary Fig. 13a). Of note, 71 genes were overlapped in these metabolic pathways of both *Slug*-deleted and aged SCs (Supplementary Fig. 13b). In addition, E2F targets and G2M checkpoints signatures being activated in *Slug*^−/− SCs were also enriched in old SCs in comparison to the young SCs (Supplementary Fig. 13c). Of those cell cycle regulators being upregulated in *Slug*^−/− SCs, 29 of which were activated in aged SCs as well (Supplementary Fig. 13d). Together, these data indicated that *Slug* insufficiency might be an important factor in causing multiple regenerative defects in aged SCs.

It has been reported that replicative senescence can play a role in the regenerative defects during normal skeletal muscle aging[32] and replicative senescence may be caused by p16^Ink4a stress pathway[33]. Next, we applied ex vivo cultured SCs as a model to

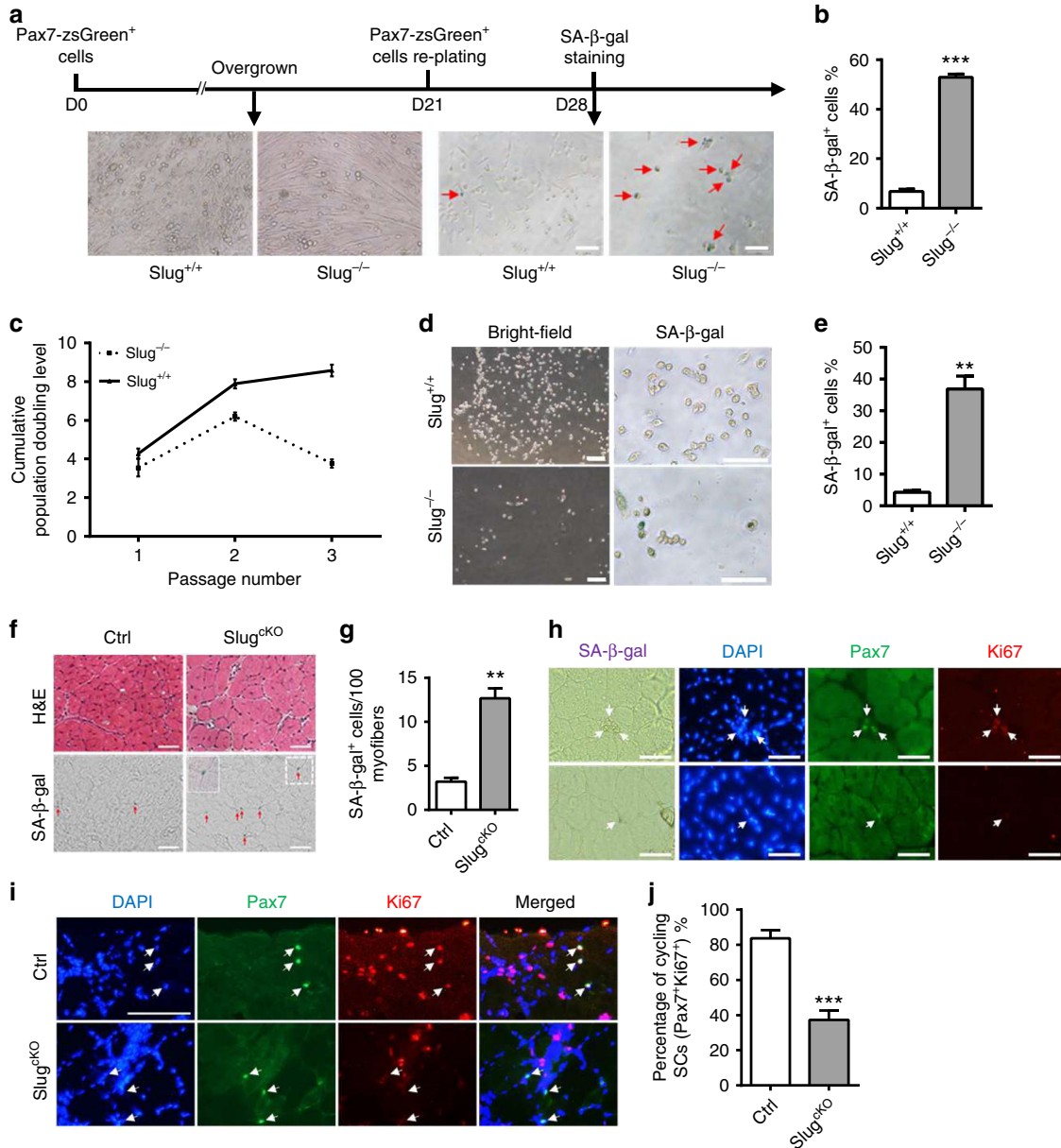

**Fig. 5** *Slug*-deficient SCs acquire senescence properties during self-renewal. **a** SA-β-Gal staining. SCs of indicated genotype were differentiated for 21 days. 5000 zsGreen⁺ reserve cells were sub-cultured for another 7 days and subject to SA-β-Gal staining. Red arrows indicate SA-β-Gal⁺ cells. Scale bar, 100 μm. **b** Quantification of the percentage of SA-β-Gal⁺ cells stained in **a**. ***$p < 0.001$ by student's *t*-test. **c** Cumulative population doubling level (CPDL) obtained in cultures of primary SC-derived myoblasts from *Slug*⁺/⁺ and *Slug*⁻/⁻ mice. **d** Photographs (left) and SA-β-Gal staining (right) of SC-derived myoblasts on day 7 of culture at passage 3. Scale bar, 200 μm (left); 100 μm (right). **e** Percentage of the SA-β-Gal⁺ cells in **d**. **$p < 0.01$ by student's *t*-test. **f** Representative images of H&E (upper) and SA-β-Gal staining (lower) on transverse TA muscle cryosections. TA muscles from *Slug*^cKO and Ctrl mice ($n = 3$ per genotype) were harvested on day 10 post injury and subject to H&E and SA-β-Gal staining. Red arrows indicated SA-β-Gal⁺ cells. The window (lower panel, right picture) represents high magnification of dotted boxed area. Scale bar, 50 μm. **g** Quantification of the SA-β-Gal⁺ cell numbers stained in **f**. **$p < 0.01$ by student's *t*-test. **h** Senescent SCs in injured mice of *Slug*^cKO and Ctrl mice. SA-β-Gal staining was combined with IHC staining against Pax7 and Ki67 on TA muscle cryosections being treated as described in **f**. Scale bar, 50 μm. **i** IHC staining for Pax7 and Ki67 in TA muscle sections harvested at day 2.5 post the second BaCl₂ injury ($n = 3$ mice per group). Arrows indicated SCs in the section. Scale bar, 100 μm. **j** Percentage of cycling (Pax7⁺Ki67⁺) SCs stained in **i**. ***$p < 0.001$ by student's *t*-test. Data are presented as mean ± SEM of three independent experiments. Also see Supplementary Fig. 12. Source data are provided as a Source Data file

address whether forced expression of *Slug* could restore the intrinsic regenerative and self-renewing capacities of the aged SCs (Fig. 7e). Similar to what found in the geriatric SCs (Fig. 7c), a sharp increase in *p16^Ink4a* expression was found in cultured SCs (Fig. 7f). Forced expression of *Slug* significantly suppressed stress-induced *p16^Ink4a* ($p < 0.001$ by student's *t*-test) (Fig. 7f). Furthermore, we demonstrated that *Slug*-overexpressing myoblasts gave

rise to a substantially larger fraction of the total MuSC population in *mdx* recipients than myoblasts transduced with the control retrovirus (Fig. 7g, h). In addition to the self-renewal capacity, *Slug* overexpression also largely restored regenerative capability of SCs that underwent passaging and subculturing for weeks, as evidenced by an increase in the number of dystrophin-positive myofibers in *mdx* recipient (Fig. 7i, j).

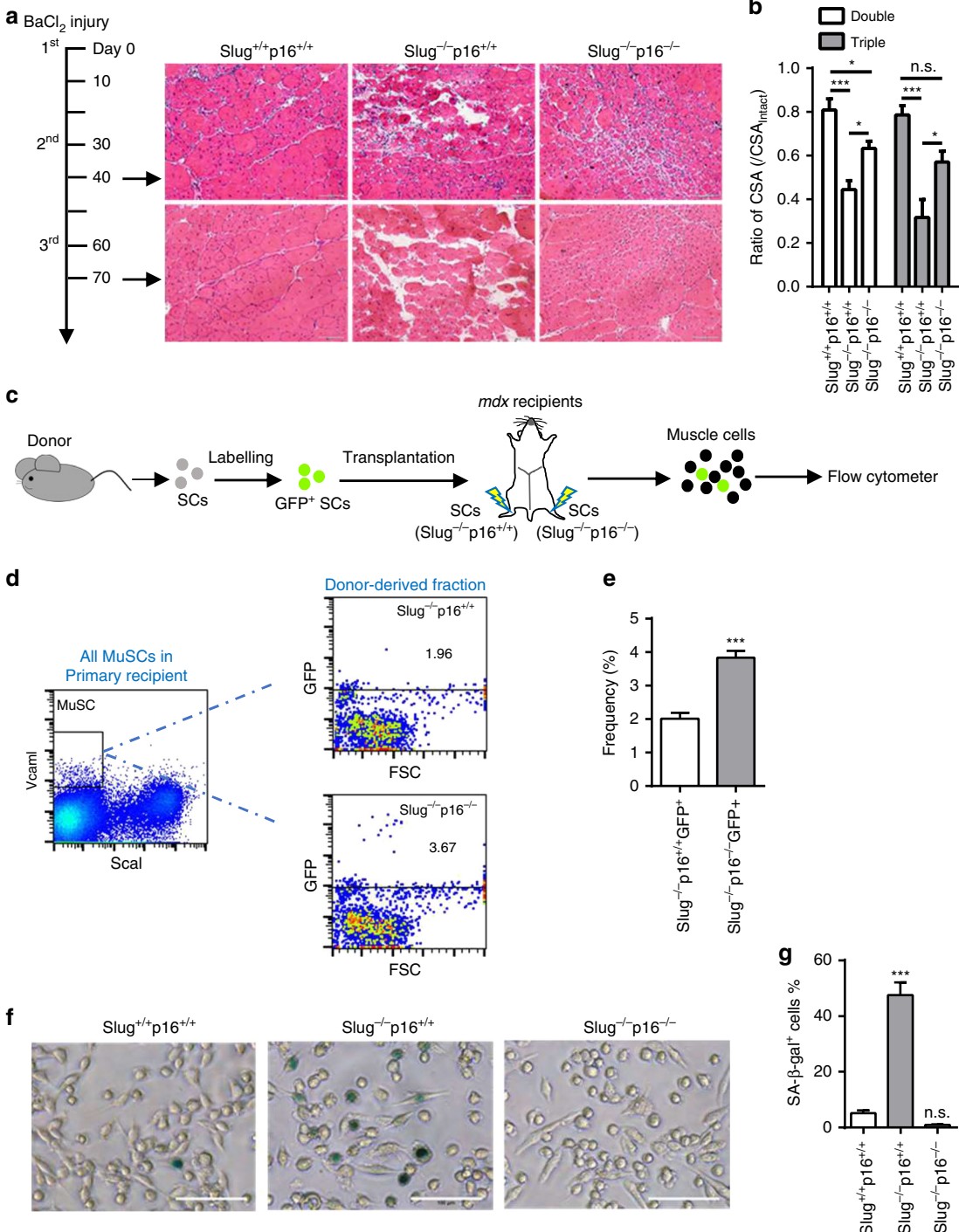

**Fig. 6** Removal of p16$^{Ink4a}$ partially rescues regeneration and self-renewal of *Slug*-deficient SCs. **a** H&E staining of the double and triple BaCl$_2$-injured TA muscles from *Slug$^{+/+}$p16$^{+/+}$*, *Slug$^{-/-}$p16$^{+/+}$*, and *Slug$^{-/-}$p16$^{-/-}$* mice ($n = 3$–6 mice per group). Left panel, the diagram for consecutive injury. Right panel, representative images of H&E staining of muscle tissue sections from injured mice. Scale bar, 100 μm. **b** Ratio of myofiber CSA in the intact and BaCl$_2$-injured TA muscles between *Slug$^{+/+}$p16$^{+/+}$*, *Slug$^{-/-}$p16$^{+/+}$*, and *Slug$^{-/-}$p16$^{-/-}$* mice. *$p < 0.05$, ***$p < 0.001$ by one-way ANOVA; n.s. not significant. **c** Scheme of SC transplantation. SCs isolated from *Slug$^{-/-}$p16$^{+/+}$* and *Slug$^{-/-}$p16$^{-/-}$* mice were infected with GFP-expressing retroviruses, and injected into either side of the BaCl$_2$-pre-injured TA muscle of *mdx* recipient, respectively. Total mononucleated muscle cells were isolated separately from either TA muscle 4 weeks after transplantation, and subjected to flow cytometric analysis for the fraction of donor-derived SCs (GFP$^+$) in MuSC (CD45$^-$CD31$^-$Sca1$^-$Vcam I$^+$) subpopulation of recipient mice. **d** Representative flow cytometry plots showing the frequency of donor-derived SCs (GFP$^+$) within the total recipient MuSC (CD45$^-$CD31$^-$Sca1$^-$Vcam I$^+$) subpopulation in mononucleated TA muscle cells. **e** Percent of donor-derived cells (GFP$^+$) in total MuSC (CD45$^-$CD31$^-$Sca1$^-$Vcam I$^+$) subpopulation of mononucleated TA muscle cells from *mdx* recipients ($n = 4$–6). ***$p < 0.001$ by student's *t*-test. **f** Representative SA-β-Gal staining of cultured primary SC-derived myoblasts. 10$^4$ SCs of indicated genotype were plated in Matrigel-coated wells of 24-well plate, and passaged weekly. Bright-field imaging and SA-β-Gal staining were performed in cells on day 7 of culture at passage 3. Scale bar, 100 μm. **g** Percentage of the SA-β-Gal$^+$ cells in serially passaged myoblasts on day 7 of culture at passage 3. ***$p < 0.01$; by student's *t*-test. n.s. not significant. Data are shown as mean ± SEM of three independent replicates. Source data are provided as a Source Data file

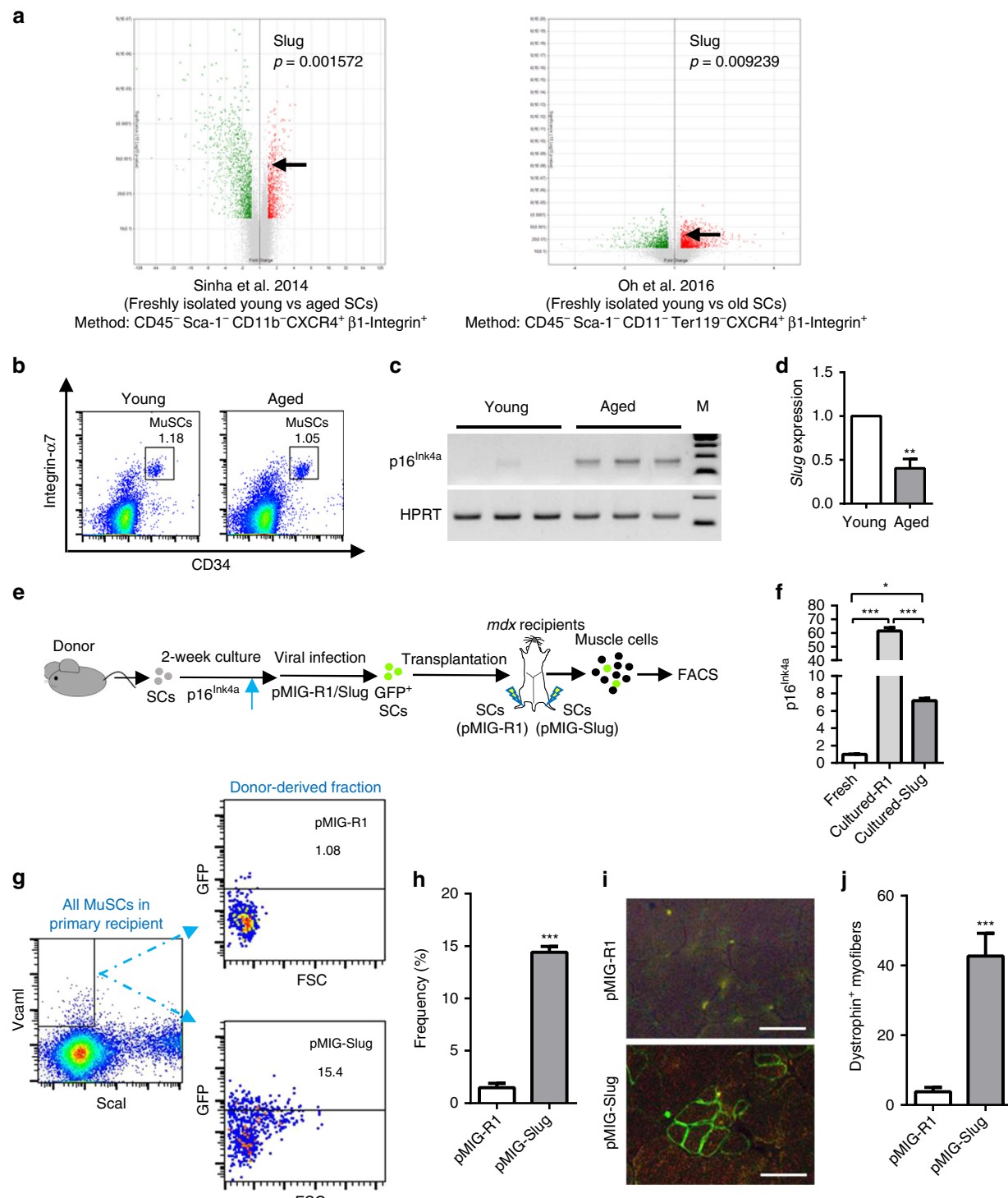

Taken as a whole, these findings demonstrated that lack of sufficient *Slug* expression is an important factor for derepression of p16[Ink4a] in aged SCs, and that restoring *Slug* expression improves p16[Ink4a]-caused regenerative and self-renewing defects in SCs.

## Discussion

Slug is a well-established regulatory transcriptional factor essential for epithelial-mesenchymal transition (EMT) during embryogenesis[34,35], tumor metastasis[36,37], adult stem cells[38–41],

and cellular reprogramming[42,43]. *Slug* is ubiquitously expressed in majority of normal tissues in the adult[13]. *Slug* knockout mice exhibit several severe abnormalities, including postnatal growth retardation[44], attenuated acute cutaneous inflammatory response[45], thinner epidermis[46], and delayed hair follicle development[47]. These defects are typical phenotypes found in mice at advanced age, reminiscent of the essential role of *Slug* in aging. Here, we described an undiscovered function of *Slug* in MuSC aging. Our results illustrated that *Slug* is highly expressed in SCs, and acts as a transcriptional repressor of *p16[Ink4a]*. Lack of *Slug*

**Fig. 7** *Slug* Overexpression Restores Repopulating Capacity of Aged SCs. **a** Volcano plots demonstrating differentially expressed genes between freshly isolated young and aged SCs from the microarray data (GEO accession: GSE50821 and GSE72179) of indicated publications. Green dots indicate substantially increased while red dots indicated substantially decreased genes in aged SCs compared with young SCs. **b** Flow cytometric plotting of MuSCs (CD45$^-$CD11b$^-$CD31$^-$Sca1$^-$Integrin-$\alpha$7$^+$CD34$^+$) in young and aged mice. **c** Qualitative RT-PCR showing up-regulation of *p16$^{Ink4a}$* in aged SCs. HPRT was used as internal control for PCR. **d** qPCR analysis of relative mRNA levels of *Slug* in SCs from the young and aged mice ($n = 3$ mice per group). **\*\****p < 0.01$ by student's *t*-test. **e** Scheme of SCs repopulation experiment. SCs were passaged weekly. After 2 weeks of culture, SC-derived myoblasts were infected with GFP-control (pMIGR1) or *Slug*-expressing (pMIG-Slug) retroviruses, and then transplanted into either side of pre-injured TA muscle of *mdx* recipient. Total mononucleated muscle cells were isolated separately from either TA muscle 4 weeks after transplantation, and subjected to flow cytometric analysis for the frequency of donor-derived SCs (GFP$^+$). **f** Quantification of relative mRNA levels of *p16$^{Ink4a}$* in quiescent and cultured primary mouse SCs with or without overexpressing *Slug*. **\****p < 0.05$, **\*\*\****p < 0.001$ by one-way ANOVA. **g** Representative flow cytometric analysis of the fraction of donor-derived SCs (pMIGR1 or pMIG-Slug) within the total recipient MuSC (CD45$^-$CD31$^-$Sca1$^-$Vcam I$^+$) subpopulation in mononucleated TA muscle cells. **h** Percent of donor-derived cells (GFP$^+$) in total recipient MuSC (CD45$^-$CD31$^-$Sca1$^-$Vcam I$^+$) subpopulation of TA mononucleated muscle cells from *mdx* recipients ($n = 3$–4). **\*\*\****p < 0.001$ by student's *t*-test. **i** Representative dystrophin immunostaining (Scale bar, 50 μm) on TA muscles transplanted with pMIGR1 or pMIGR1-Slug retrovirus-infected myoblasts. **j** Quantification of dystrophin-expressing myofibers in TA muscles of recipient *mdx* mice ($n = 6$ mice per group) shown in **i**. **\*\*\****p < 0.001$, Student's *t*-test. Data are shown as mean ± SEM of three independent replicates. Unprocessed gel blots are provided in the Supplementary Fig. 14 and source data file. Also see Supplementary Fig. 13. Source data are provided as a Source Data file

triggers SCs into pre-senescence state by accumulating p16$^{Ink4a}$ expression and compromising SC self-renewing capacity after activation, leading to MuSC exhaustion and muscle regeneration defect. This study revealed a cell-autonomous mechanism of SCs losing their intrinsic regenerative and self-renewal capacities during chronological aging.

The role of *Slug* in skeletal muscle regeneration was firstly reported by the Hoffman group and they showed that *Slug* is a downstream target of MyoD and its expression was markedly increased at the whole muscle tissue level on day 4 and 10 during muscle regeneration[33]. Of note, they demonstrated that *Slug* knockout mice were defective for muscle regeneration after the first round of injury by cardiotoxin, which is different from our current findings in the global and conditional *Slug* knockout mouse models. Here, we demonstrated that *Slug* is highly expressed in quiescent SCs and its expression is greatly reduced in differentiated myotubes. This is consistent with the more recent work showing gradually diminished Slug protein in differentiating myoblasts in vitro[22]. Our current findings do not support that *Slug* is a downstream target of MyoD, since *Slug* expression is abundant in MyoD-negative quiescent SCs. As revealed by Soleimani et al., Snai1/Slug compete with MyoD and prevent its occupancy on differentiation-specific regulatory elements in undifferentiated primary myoblasts[22]. Therefore, lack of competitive Slug binding should facilitate easier access of MyoD to those differentiation-regulating elements on its first appearance after injury and initiate myogenic program. Indeed, *Slug*-deficient mice were able to regenerate their muscle upon first round of muscle injury. In our current studies, we further proved that the failure of muscle repair in *Slug*-deficient mice after multiple injuries was partially due to the exhaustion of SC pool that was caused by their SC self-renewing defect. However, in view of the abundant expression of *Slug* in myoblasts, it is possible that *Slug* is playing a role outside of self-renewal, which could affect muscle regeneration and transplantation potential, by biasing SC proliferation versus differentiation.

In our current study, *Slug* deletion caused only a small increase of *p16$^{Ink4a}$* mRNA in quiescent SCs and did not prevent their activation and differentiation following the first muscle damage. We infer that the compensatory regulation of Snai1, another member of the Snail/Slug superfamily of zinc-finger transcription factor, was probably also crucial to keep *p16$^{Ink4a}$* transcription under control. This conjecture is supported by the fact that both Snai1 and Slug recognize and bind to the identical E-box motifs in myoblasts[22]. Interestingly, the Gridley group have reported a compensatory regulation of the *Snail1* and *Snail2* (*Slug*) during chondrogenesis[48]. Such compensation from Snai1 expression

might also facilitate muscle regeneration as observed in *Slug*-deficient mice after first round of muscle injury as Snai1 can indirectly influence key myogenic transcription factors like Myf5 to activate myogenesis[49]. Of note, both Snai1 and Slug proteins decrease during myoblast differentiation[22]. Therefore, we reasoned that the large increase in *p16$^{Ink4a}$* mRNA expression in cultured *Slug*-deficient myoblasts and accumulated p16$^{Ink4a}$ protein in activated SCs were due to the loss of the dual regulation conferred by both Slug and Snail1. Future studies using *Snai1* and *Slug* double knockout mice will be essential in testing this hypothesis. In addition, an increase in *p16$^{Ink4a}$* mRNA but not protein expression level in quiescent *Slug$^{-/-}$* SCs suggests a post-translational regulation mechanism. It is possible that p16$^{Ink4a}$ protein is unstable in resting SCs. Indeed, it was reported that p16$^{Ink4a}$ translation is suppressed by miR-24[50], which is highly expressed in quiescent SCs but greatly downregulated in activated SCs[51]. These findings might explain our finding that *p16$^{Ink4a}$* mRNA but not protein accumulated in quiescent *Slug$^{-/-}$* SCs.

Quiescent SCs rely on fatty acid and pyruvate oxidation to maintain a low metabolic rate in quiescence. The energy supply switches to glycolysis during early activation[52], while both mitochondrial density and oxidative phosphorylation activity are increased following differentiation[53–55]. It was previously demonstrated that the mitochondrial-associated metabolism pathway is more silent in Pax7$^{Hi}$ SCs being of higher level of stemness and responsible for self-renewal[56]. A more recent study of in vitro culture of SCs in conditioned medium favoring either glycolysis or oxidative phosphorylation showed that a shift from glycolysis to oxidative phosphorylation negatively affects the return to quiescence of activated SCs[57]. Although SCs remain in quiescence in the absence of *Slug*, a batch of genes in glycolysis, oxidative phosphorylation, and nutrient-sensitive PI3K-AKT-mTOR pathways were upregulated therein. Such metabolic gene expression profiling is consistent with our observation of the attenuated self-renewal and accelerated senescence in *Slug* null SCs.

An aging-associated reduction of *Slug/Snai2* expression was observed in mouse SCs. Mechanistically, *Slug* expression was reported to be under control of a number of signaling pathways such as FGF, Wnt, TGFβ, Notch, stem cell factor (SCF), integrins, and estrogens etc.[58]. Several of these signaling molecules including FGF[3] and Notch[59,60] are known as inducers of *Slug* and were reported to decline with age in mouse SCs. Therefore, we speculate that these impaired upstream signaling pathways might account for *Slug* insufficiency in aged SCs. Interestingly, active Notch and Notch ligand Delta are also decreased in old human muscle when compared to young

muscle[61]. Indeed, we identified the potential Slug-binding consensus sequence (E-box) in the promoter region of human *p16Ink4a* gene as well (Supplementary Fig. 9c). Gain and loss of function of SLUG in primary human myoblasts causes down and upregulation of *p16INK4A* expression, respectively (Supplementary Fig. 9d-g). Function of *Slug* in SCs might therefore be highly conserved in mouse and human. Future studies exploring small molecules that are able to induce *Slug* expression in aged mice are warranted to test the improvement of aging-associated muscle stem cell defects in vivo.

In conclusion, although p16Ink4a has been well recognized as a key factor for stem cell aging, upstream regulators for causing its dysregulation remain unknown. Our studies demonstrated that Slug plays an important role in repressing *p16Ink4a* transcription in skeletal MuSCs. Loss of *Slug* gives rise to severe regenerative defects during continuous muscle injury by accelerating entry of SC senescence. Restoration of *Slug* expression during chronological aging rejuvenates the function of aged SCs by suppressing *p16Ink4a* expression. These results offer a promising therapeutic target for aging-associated degenerative muscle disease.

## Methods

**Mice**. C57BL/6, C57BL/6-Tg(CAG-EGFP)10sb/J, Pax7tm1(cre)Mrc/J, B6.Cg-Pax7tm1(cre/ERT2)Gaka/J, and C57BL/10ScSn-Dmdmdx/J mice were purchased from Jackson Laboratories. p16Ink4a knockout mice were obtained from the NCI Mouse Repository. Pax7-zsGreen transgenic mice were a kind gift from Dr. M. Kyba (University of Minnesota). Slug conditional knockout mouse line was generated in the current study. All compound genetically-engineered mice were a result of breeding the above strains followed by appropriate PCR-based genotyping. From 4 to 8-week-old male mice were used as adult mice in all the experiments, while 2-year-old male mice were used as aged mice. All the animal studies were approved by the Animal Care and Use Committee (approved protocol number: 16–111) and performed in compliance with the institutional guidelines of the University of Illinois at Chicago.

**Gene targeting and generation of a *Slug*-floxed allele**. The *Slug* targeting construct was generated by PCR using 129X1/svJ mouse genomic DNA. Primers for generating the targeting construct are shown in Supplementary Table 1. First, a ~3 Kb genomic sequence containing *Slug* promoter region was PCR-amplified as the left arm with SlugKO-P1 and SlugKO-P2 primers and cloned into NotI and SalI sites of pLoxP-2FRT vector (designated as pLoxP-2FRT_SlugN). Second, a ~4 Kb genomic sequence covering the genomic region from the immediate downstream of SlugKO-P2 binding site to the intron 3 was PCR-amplified as the right arm with SlugKO-P3 and SlugKO-P4 primers, followed by cloning into BamH1 and Kpn1 sites of pBS vector (designated as pBS_SlugC). Third, a LoxP linker was cloned into EcoR1 site of pBS_SlugC and the resultant plasmid was named as pBS_SlugC-LoxP. Lastly, the SlugC-LoxP fragment was isolated from pBS_SlugC-LoxP with BamH1 and Kpn1 and then cloned into pLoxP-2FRT_SlugN. The resulting construct was designated as p2LoxP-2FRT_Slug, which carries neomycin (neo) for positive selection and HSV-tk marker for negative selection. The linearized targeting construct was nucleofected into R1 ES cells (129/svx129/Sv-CP F1) by Nucleofector II (Lonza, Allendale, NJ), and followed by positive selection with G418 and negative selection with FIAU. Correctly targeted ES cells were first screened for the presence of the DNA sequence containing LoxP and FRT sites by PCR with SlugKO-P11 and Neo-R1 primers, then screened for the left arm by long-PCR with SlugKO-P10 and Neo-R1 and confirmed by DNA sequencing. Two positive ES cell clones were expanded and microinjected into C57BL/6 blastocysts to generate chimeric Slugflox-neo/+ founder mice at the Mouse Transgenic & Gene Targeting Core of Maine Medical Center Research Institute. The neo cassette was deleted from the germline by crossing Slugflox-neo/+ mice to Flp-expressing transgenic mice (ACTB:FLPe B6; SJL, The Jackson Laboratory), which express Flp recombinase under the control of the human *actin* promoter[62]. The *Flp* gene was subsequently removed by backcrossing with C57BL/6 mice to generate Slugfl/+ mice. The Slugfl/+ allele genotyping was performed by PCR with SlugPT-F and SlugEX1-R primers (Supplementary Table 1).

**Generation of SC-specific *Slug* knockout Mouse Line**. Female Slugfl/fl mice were bred with male mice (Pax7tm1(cre)Mrc/J) homozygous for the *Cre*-recombinase gene under the control of endogenous Pax7-promoter to generate SC-specific *Slug* knockout mouse line (Slugfl/flPax7Cre/+ or Slugfl/flPax7CreER). Genotyping for *Cre* gene was performed using a multiplex PCR method according to the provided protocol. To delete *Slug* in adult SCs, tamoxifen (Sigma) was administered intraperitoneally (100 mg kg⁻¹ body weight per day) for 5 consecutive days.

**Muscle stem cell isolation**. Murine hindlimb skeletal muscles were dissected, cleaned by removal of non-muscle tissues and subjected to a gentle collagenase (100 U ml⁻¹, Catalog no. 171001015, Life technologies, Carlsbad) and dispase (2 U ml⁻¹, Catalog no. 17105041, Life technologies, Carlsbad) digestion in DMEM at 37 °C water bath for 1 h. Enzymatically digested muscle mass was re-suspended in DMEM with 10% horse serum, and further mechanically triturated through vigorous and consecutive passing through a 10-ml glass pipette and subsequently 9" cotton-plugged glass Pasteur pipette. Resultant cell suspension was passed through sequential 70-μm and 40-μm strainer (BD Bioscience, San Jose, USA) to generate bulk skeletal muscle single-cell suspensions.

To isolate MuSCs from undamaged muscles, mononucleated cells were stained with biotinylated antibodies reactive to CD45, CD11b, CD31, and Sca1, and then incubated with streptavidin-APC-Cy7, integrin-α7 antibody labeled with PE (MBL, Woburn, USA), and eFluor660-conjugated CD34 antibody (eBioscience, San Diego, USA). Cells negative for CD45, CD31, CD11b, and Sca1 but positive for CD34 and integrin-α7 were sorted as MuSCs by fluorescence-activated cell sorting (FACS)[63].

To isolate MuSCs from damaged muscles, mononucleated cells were first stained with biotinylated antibodies reactive to CD31 and CD45, followed by incubation with streptavidin-APC-Cy7, Sca1 antibody labeled with FITC (eBioscience, San Diego, USA), and PE-conjugated Vcam I antibody (Miltenyi Biotec, Bergisch Gladbach, Germany). Cells negative for CD31, CD45, and Sca1 but positive for Vcam I were sorted as MuSCs by FACS[19].

To isolated MuSCs from Pax7-zsGreen transgenic mice, zsGreen⁺ cells were directly sorted from mononucleated cells by FACS.

**Myofiber culture and AraC treatment**. EDL muscles were dissected and digested for 1 h in DMEM with 0.2% Collagenase type I. Pre-digested EDLs were then transferred to horse serum-precoated dish containing prewarmed DMEM, and gently flushed with a fire polished large bore glass pipette to release myofibers. Live and healthy single myofibers were obtained by three consecutive transfers of individual myofibers with small bore glass pipette into new horse serum-precoated dishes containing prewarmed DMEM. Live myofibers were cultured in uncoated plastic 24-well-plates at density of 1 fiber per well with DMEM supplemented with 15% horse serum and 1 nM bFGF (PROSPEC, East Brunswick, USA). For AraC experiment, myofibers were cultured for 72 h and then incubated with or without 100-μM AraC (C1768, Sigma, USA) for 48 h (from day 3–5) and fixed (day 5) for immunostaining.

**Skeletal muscle cell culture and differentiation**. Mouse skeletal MuSCs were cultured using the MuSC growth medium [DMEM/F10 (1:1) with 20% FBS, 2.5 ng ml⁻¹ bFGF (PROSPEC, East Brunswick, USA) and 1% penicillin/streptomycin] in Matrigel (BD Biosciences, San Jose, USA) precoated tissue culture plates. To induce differentiation, cells were switched from growth medium into the fusion medium (DMEM supplemented with 5% horse serum) for 5 days.

**Comparison of proliferation potential**. Cumulative population doubling level (CPDL) was calculated using the formula $\chi = [\log_{10}(N_H) - \log_{10}(N_1)]/\log_{10}(2)$[64] in which $N_1$ is the inoculum cell number and $N_H$ is the cell harvest number. To determine cumulated doubling level, population doubling level for each passage was calculated and then added to the levels of the previous passages. Cumulative doubling number was first calculated from passage 1.

**Histology and immunohistochemistry in muscle cryosections**. Fresh TA muscles were embedded in Tissue-Tek® O.C.TTM compound (Fisher Scientific, Hampton, USA), frozen in liquid nitrogen-cooled isopentane, and stored at −80 °C until analysis. Frozen muscles were cross-sectioned (10 μm) using a Leica CM1850 cryostat. For histology, the sections were dried at room temperature for over 3 h before staining. Sections were then rehydrated in PBS for 5 min and fixed in 10% formalin for 15 min, and proceeded with routine haematoxylin and eoxin (H&E) staining. For immunohistochemistry (IHC) study, air-dried muscle sections were fixed with 4% paraformaldehyde (PFA) and permeabilized in 0.25% Trition X-100. Tissue sections were then blocked in phosphate buffer saline (PBS) with 5% goat serum, 2% bovine serum albumin, and 1% Tween- 20 for 1h, and then followed by incubation with anti-Pax7 (DSHB, Iowa, USA, 1:10), anti-Laminin (clone A5, Invitrogen, Carlsbad, USA, 1:100 dilution), anti-p16 (M-156, Santa Cruz Biotechnology, USA), anti-Ki67 (Clone SolA15, eBioscience, USA, 1:100), and/or anti-dystrophin (Clone MANDYS8, Sigma, USA, 1:100) primary antibodies overnight at 4 °C. After washing with PBS, sections were then incubated with Alexa-conjugated secondary antibodies (Invitrogen, Carlsbad, USA, 1:200 dilution).

**Immunofluorescence staining**. Freshly isolated SCs were cytospun onto slides and fixed with 4% PFA. After being permeabilized by 0.2% Trition X-100, cells were then blocked in PBS with 1% bovine serum albumin (BSA) for 1 h at room temperature followed by staining with anti-Pax7 (DSHB, Iowa, USA, 1:10) and anti-Slug (Clone C19G7, Cell Signaling, USA, 1:400) primary antibodies overnight at 4 °C. Cells were washed three times with PBS and incubated with appropriate secondary antibody for 1 h at room temperature.

**SA-β-galactosidase staining**. Cellular senescence was evaluated by β-galactosidase activity using MarkerGene™ Cellular Senescence Assay Kit (Marker Gene, Eugene OR, USA). Briefly, cells or tissue sections were fixed for 4 and 20 min, respectively. Fixed samples were then washed in PBS (pH7.0) twice for 10 min and incubated with X-gal containing staining buffer at 37 °C overnight for cells and 48 h for tissue sections. For tissue sections, X-gal substrate was changed after 24 h. Samples were washed in PBS, and post-fixed in 1% PFA 5 min for cells and 30 min for sections. After being washed three times for 10 min in PBS, samples were mounted in PBS, 20% glycerol or proceeded for IHC.

**Muscle injury and SC transplantation**. Experimental mice were anesthetized by intraperitoneal injection of ketamine (100 mg kg$^{-1}$ body weight) and xylazine (10 mg kg$^{-1}$ body weight). For single muscle injury, TA muscles were injected with 50 μl of 1.2% BaCl$_2$ (B0750, Sigma, USA) at multiple sites and harvested 10 days post injury. For double or triple injuries, mice were allowed to recover for 1 month after last BaCl$_2$ injection, and then injured again with 50 μl of 1.2% BaCl$_2$. For in vivo cellular senescence detection, TA muscles were injured with 50 μl cardiotoxin (*Naja pallida*, 10 μM; Calbiochem, USA). For SC transplantation, recipient *mdx* mice were anesthetized by intraperitoneal injection of ketamine (100 mg kg$^{-1}$ body weight) and xylazine (10 mg kg$^{-1}$ body weight). TA muscles were injected with 50 μl of 1.2% BaCl$_2$ 1 day before transplantation. Next day, SCs (or myoblasts) were re-suspended in 15 μl of PBS and then injected into pre-injured TA muscle. TA muscles were harvested from recipient mice 4 weeks after transplantation and analyzed by cryosectioning and microscopy.

**RNA extraction and qPCR**. Total RNA was extracted from samples (SCs or myoblasts) using Quick-RNA™ MicroPrep kit (ZYMO Research, USA). First strand cDNA synthesis was performed using RT™ Master Mix (LAMDA BIO-TECH). RT-PCR was performed using GoTaq® Green Master Mix (Promega, Madison, USA) on Bio-Rad Thermal Cycler. qPCR was performed using Bullseye EvaGreen Qpcr MasterMix (MIDSCI, USA). The primer sequences for PCR are listed in Supplementary Table 1.

**cDNA microarray and data analysis**. Total RNAs extracted using Quick RNA™ MicroPrep kit (ZYMO Research, Irvine, USA) were first analyzed by Bioanalyzer (Agilent Technologies, Santa, Clara, USA). A triplicate of RNA samples with an RNA integrity number (RIN) >9.0 were used for subsequent labeling and hybridization with Mouse Gene 2.0 ST Arrays (Affymetrix, USA). Expression data were processed using Gene Expression Console software (Affymetrix, Santa Clara, USA). The significance of differentially expressed genes was determined using Transcriptome Analysis Console software (Affymetrix, Santa Clara, USA). GSEA version 3.0 (http://software.broadinstitute.org/gsea) and MSigDB gene sets version 6.1 were used to determine the sets of genes that are statistically different between two groups using GSEA preranked mode[25]. Enrichment Map[65,66] was used to visualize enriched pathways. GSEA output files were subjected to the Enrichment Map app with the following cutoff parameters to build the map: p value < 0.001, FDR q-value < 0.05 and overlap similarity coefficient >0.5. Nodes in the Enrichment Map represent a set of genes and their connections in which the set of genes have in common nodes. The edge thickness is proportional to the overlap of two gene sets. The node colors map enrichment significance: blue/downregulated, red/upregulated. Gene ontology analysis (GO) on retrieved RNA-seq dataset (GSM937218) was conducted in the Database for Annotation, Visualization, and Integrated Discovery v6.7 (DAVID, http://david.abcc.ncifcrf.gov) using all detectable genes as the background. Only enriched pathways with FDR value < 0.05 are presented.

**Retroviral transduction of myogenic progenitor cells**. Two hundred ninety-three T cells were transfected with a mixture of DNA plasmids containing 1.5 μg of Pcl-Eco (IMGENEX), 2.5 μg of pMIGR1 vector or pMIGR1 harboring Slug by Fugene HD (Roche) according to the manufacturer's instruction. Virus-containing media were collected 24 h after transfection and filtered through a 0.45-μm pore-size filter. SCs were transduced with retrovirus-containing media supplemented with 4-μg ml$^{-1}$ polybrene (Sigma), followed by centrifugation at 1000 × g for 1 h.

**ChIP assay**. ChIP assay was performed with ChIP-IT Express Enzymatic kit (Active Motif, Carlsbad, CA, USA) according to the manufacturer's instruction. Genomic DNA was pulled down by anti-Flag (FG4R, Thermo Scientific, USA), anti-HA (Thermo Scientific, USA), or anti-IgG (Millipore, USA) control antibodies. Primers for amplifying promoter region of p16$^{Ink4a}$ from immunoprecipitated DNA fragments are listed in Supplementary Table 1.

**Tamoxifen induced SC-conditional *Slug* knockout**. Experimental mice were intraperitoneally injected with tamoxifen (100 mg kg$^{-1}$ per day) for 5 days. First BaCl$_2$ injury was conducted in TA muscles 5 days after the last injection of tamoxifen. For double injury, mice were recovered for 1 month after the first injury, and subject to a second BaCl$_2$ injection at the same sites. TA muscles were harvested on day 10 post each injury.

**Epitope Flag/HATagging by CRISPR/Cas9-based genome editing**. A gRNA targeting a preselected site near the stop code (TGA) in the exon 3 of *Slug* gene was designed (left panel) and cloned into pAAV-gRNA-*GFP* expression vector. The donor template plasmid contains a DNA sequence encoding three copies of HA and Flag tags and blasticidin resistance gene (Blast), which is flanked by the two-homology left and a right arm sequence. The blasticidin sequence is in-frame fused to the FLAG tag via a P2A sequence. The three copies of HA tag is in-frame fused to the end of *Slug* (right before stop code). The PAM sequence (**TGG**) was changed to "**TaG**" in the left arm of the donor vector to avoid binding by the gRNA, but without altering protein sequence (right panel). Myoblasts were co-transfected with Cas9 and/or the gRNA expression vectors, together with the donor template plasmid. Three days later, blasticidin was added into culture medium to selected cells with correct site-directed integration of the donor sequence (three copies of HA and FLAG tags and Blast).

**Luciferase reporter assay**. Myoblast cells were seed in 48-well plates and cultured in myoblast growth medium. A total of 0.5 μg of DNA containing 0.1 μg of PGl3-p16 luciferase reporter plasmid, 0.4 μg of expression plasmid (pMIG-R1 or pMIG-Slug) was transfected using PEI. A co-transfected 0.025μg CMV-lacZ plasmid was used to normalize transfection efficiency. Cells were lysed 24 h using Glo Lysis Buffer (E266A, Promega) after transfection, and the luciferase activity of each extract was assayed using Bright-Glo™ Luciferase Assay System (E2620, Promega). One microliter cell lysate was added into 399-μl Z buffer. One hundred microliter ONPG (o-nitrophenyl b-D-galactopyranoside; Sigma #N-1127) with concentration of 4 mg/ml was added and mixed well. When the sample become yellow, 500 μl 1M carbonate was added to stop the reaction and the activity of beta-galactosidase was measured at A420 to normalize the luciferase activity.

**Western blot analysis**. Total protein was lysed using CelLytic™ MT Cell Lysis reagent (#C3228, Sigma, USA) supplemented with protease inhibitor cocktail (#P8340, Sigma, USA). Cell lysates were centrifuged for 15 min at 13,000 × g, and supernatants were collected for protein concentration determination using Pierce™ BCA Protein Assay kit (#23225, ThermoFisher Scientific). Fifteen micrograms of total protein from each sample were resolved by 10% SDS-PAGE and electroblotted onto polyvinylidene difluoride membranes. Immunoblotting was performed with anti-Flag (FG4R, Thermo Scientific, USA), anti-HA (5B1D10, Thermo Scientific, USA), anti-β-actin (Cell Signaling Technology #4970, USA), and anti-Slug (#9585, Cell Signaling Technology, USA) antibodies at 4 °C overnight. After washing, membranes were incubated with a horseradish peroxidase-conjugated goat anti-rabbit IgG antibody (#1705046, Bio-Rad, USA) for 1 h at room temperature. Bound antibodies were detected using Clarity Max™ Western ECL substrate (#68SC, Bio-Rad, USA). Uncropped blots can be found in Supplementary Fig 14 and the Source data file.

## Data availability

Data supporting the findings of this manuscript are available within the article and its supplementary information files or from the corresponding author upon reasonable request.

All data from DNA microarray experiments have been deposited in Gene Expression Omnibus database under accession code GSE128507.

The source data underlying Fig. 1–7 and Supplementary Figs 1, 3-5, 8-10 and 12 are provided as a Source Data file.

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

## Acknowledgements

We thank Dr. Liangyou Rui at The University of Michigan for his guidance on the generation of *Slug* conditional mouse line and providing mouse R1 ES cells and pLoxP-2FRT vector. We also thank Dr. Michael Kyba for provision of the Pax7-szGreen mouse strain. This work was funded by NIA R01-AG040182 to W.S.W.

## Author contributions

Conceptualization, P.Z. and W.S.W; Methodology, P.Z., C.P., Y.G., F.W. and Y.Z.; Investigation, P.Z., C.P., Y.G., F.W. and Y.Z.; Formal Analysis, P.Z., C.P., Y.G., F.W. and Y.Z.; Writing, P.Z. and W.S.W.; Resources, P.Z., Y.G., F.W. and Y.Z.; Funding Acquisition, W.S.W; Supervision, W.S.W.

## Additional information

**Competing interests:** The authors declare no competing interests.

