## [Peer Review File · Nature Communications]

Reviewers' Comments:

Reviewer #1:

Remarks to the Author:

The authors show that slug plays a role for muscle regeneration through regulating p16. They show loss of slug in SCs leads to p16 upregulation and induction of senescence in a p16 dependent manner. Consistently, the loss of slug in SCs results in decreased regenerative capacity during repeated injury. In addition, this is partially rescued by p16 ablation as well as enforced expression of slug. Data suggest a new link between slug and p16/senescence and its key role in muscle regeneration. The experiments are generally well designed.

Sup fig. 1d shows a reduced myofiber size in whole body slug KO mice, but this doesn't seem to be the case in cKO? The authors might want to comment on this.

Fig. 3g/h. This is a nice experiment but not quantitative. The figure only shows a single result. They need to show reproducibility of the result.

Fig. 4c. please provide normalized enrichment scores (NES) and adj-p values.

Fig. 4g. This is not quantitative and they need to show reproducibility. It is also advisable to include negative control regions. Are these cells 'primary' or (somehow) immortalised myoblasts? To conclude that "Slug is a transcriptional repressor for p16Ink4a", they need to provide functional validation using reporter assay with the wt/mutant promoter, otherwise tone down.

Fig. 4h: it would be more informative if the authors show the difference between wt resting and wt cultured. Using the ChIP assay, can they see the reduction of slug on the p16 promoter during culture?

Fig. 4i,j: "these results indicated that Slug deficiency leads to an increase in p16Ink4a transcription in SCs, and replicative stress signaling triggered by SC activation and proliferation concurrently increases the stability of p16Ink4a mRNA". This is not supported by the data. Actually, the logic doesn't make sense. How can one hypothesise that mRNA is unstable when it is high. It would make more sense to speculate that p16 protein is unstable in resting SCs?

Fig. 5c: cumulative PD is decreased between p2 and 3, suggesting a substantial number of cells might undergo cell death (or plating efficiency becomes low). The authors need to clarify whether slug ko induces both senescence and cell death thus residual (survival) cells are used for the assays in vitro.

"..., muscle repair was significantly improved in Slug^{-/-}-p16^{-/-} mice after injuries (Fig. 6a,b)". 'Statistical significance' is not shown.

Minor points:

Fig. 4f seems to show tagging of p16, not slug?

Fig. 7a: which dots represent slug?

Reviewer #2:

Remarks to the Author:

In general this is a well written manuscript with significant evidence provided that the EMT transcription factor Slug/Snai2 can control muscle stem cell quiescence/senescence through the repression of p16Ink4a . The SC-specific deletion of a novel conditional Slug allele using Pax-7 Cre

mouse line is very convincing concerning the cell-autonomous role of Slug in regulating SC quiescence/senescence. In addition, the rescue of muscle regeneration of Slug deficient mice by breeding onto p16Ink4a deficient background provides very strong genetic evidence of the important of p16 in the Slug null phenotype.

Major concerns:

- 1) The authors show that there is a difference in Snai2 expression in SC in the muscle between young and old mice and that this correlates with changes in p16 expression. However, no mechanistic insight or discussion is provided concerning exactly how or why Snai2 levels fail off over time in the SC population. Is a similar correlation observed in human muscle samples in young vs old or in control vs diseased muscle biopsies? From a regenerative medicine perspective, how would one potentially modulate Snai1 expression in the adult?
- 2) Given the gene changes that occur in Snai2 null SC populations that are involved in cytokinesis, chromosome segregation and microtubule assembly how sure are the authors that asymmetric vs symmetric cell division is not affected and contributing to the regeneration defect given the importance of asymmetric self-renewal in the SC cells of the muscle (see Kuang et al., Cell 2007).

Minor issues:

- 1) In the discussion the authors mention the possibility of Snai1 partial compensation in Snai2 SC phenotype but do not mention recent paper by Sieiro et al., eLife 2016 whereby it was demonstrated that Snai1 levels can indirectly influence important muscle transcription factors such as Myf5.
- 2) Figure 4F is confusing and needs to be changed. It is Snai2 locus that has been FLAG tagged at C-terminus and not p16Ink4a. This should be removed as it is depicted in the Sup. Fig 3? In this Figure primers should be indicated that flank E box element. How conserved is this E-box element? Is this E-box element present in the human p16Ink4a promoter (see major concern 1 above).
- 3) How sure are the authors that the C-terminal tag on SNAI2 that was used for CHIP experiments does not interfere with the normal function of SNAI2? How does this compare to N-terminal tags? Would be good to show that protein half life and localization is not affected by this Tag.

Reviewer #3:

Remarks to the Author:

Review of Zhu et al., 2018 Nature Communications manuscript

The manuscript entitled "Transcription factor Slug/Snai2 reinforces self-renewal in aged skeletal muscle stem cell" by Zhu et al., (2018) attempt to demonstrate that the zinc-finger transcription factor Slug is highly expressed in quiescent muscle stem cells, Satellite Cells (SCs), and function as a direct transcriptional repressor of the cell cycle inhibitor, p16Ink4a. Then, loss of Slug promotes increases expression of p16Ink4a in SCs and accelerates the entry of SCs into a senescent state upon damage-induced stress. Therefore, p16Ink4a depletion partially rescues defects of Slug null SCs. The study also, through the body of the results section, infers that decreased level of Slug expression and consequent de-repression of p16Ink4a in muscle SCs during aging, intrinsically impact their self-renewal potential after multiple round of regeneration. It should be noted that the authors of the manuscript under review have previously shown that Slug repress self-renewal of hematopoietic stem cells (HSCs) and is essential for controlling the transition of HSCs from relative quiescence under steady-state condition to rapid proliferation under stress conditions (Sun et al, 2010).

In summary, the manuscript by Zhu et al, (2016) has identified of a novel upstream regulator of p16Ink4a. However numerous technical flaws minimize the authors conclusions

General points:

First, all the experiments are conducted with Slug germline knockout or Pax7-Cre conditional knockout mice, raising the possibility that the phenotype is not confined only to muscle SCs, calling into question the proposed Slug intrinsic role in SCs. Also, the phenotype can derive from post-natal maturation mechanisms that influence adult homeostasis, proliferation, differentiation and self-renewal potential of SCs, during muscle regeneration. So, to this point, it is essential to use Pax7CreER tamoxifen inducible mice models to determine the specific functions of Slug solely in quiescent SCs and their downstream progeny during muscle repair.

Second, to determine a possible SCs self-renewal potential defects, the author take advantage of transplantation assay and quantify self-renew SCs mainly by FACS analysis of VCAM+/GFP+ SCs from WT or Null mice. As they state in the text, one-month post transplantation some of the donor SCs may be still proliferating, therefore the use of VCAM marker (VCAM marks SCs and their progenitors) cannot be used to determine the fraction of self-renewed SCs. There are two important criteria that SCs must fulfill to be considered 'self-renewed': 1) location: SCs must re-enter the muscle fiber niche and reside under the basal lamina; 2) quiescent state: upon muscle injury SCs activate and acquire proliferation markers (Ki67+/MyoD+). Once they self-renew, they must go back to quiescence (Ki67-/MyoD-). Therefore, the authors need to perform more detailed experiments to determine if Slug impact SCs self-renewal.

Third, the authors need to perform a more careful analysis of cell fate during the repair process- while they argue for a senescence mediated control of self-renewal, in vitro experiments suggest that a large number of progenitors undergo senescence-not the cells that return to quiescence. If a large number of progenitors are senescing in vitro-why does this not manifest as a phenotype in vivo? Is this due to the use of germline mouse in senescence experiments, or a decline in the number of myonuclei during regeneration that was not scored by the authors.

Fourth, based on their title a reader would expect more figures related to SCs self-renewal decline during aging. Only the final figure attempt to connect Slug, self-renewal and aging. During aging, the number and function of muscle SCs decline. Recently, it has been showed also that maintenance of quiescence in adult mouse life relies on active repression of senescence pathway (Sousa-Victor P et al., 2014), mainly through p16Ink4a. Therefore, does loss of Slug in young cells make them like aged/geriatric SCs?

Detailed points:

Figure 2A: Rather than having SCs compared to DN cells, which are almost unknown population, it will be more informative and relevant to compare population previously characterized such as FAPS, CD31 endothelial cells (where the role of Slug is well characterized) or blood CD45+ cells.

Figure 2B: Please quantify slug expression in activated SCs, for example 2-3 days in proliferation conditions. If Slug is increased in proliferating cells, how would this change the authors conclusions?

Figure 2F: Please quantify Slug+ Pax7+SCs; quantification of Pax7+ SCs need to be more accurate: please stain adult tissue section from Ctrl and Slug cKO mice for Pax7, Laminin and Ki67/MyoD.

Figure 2: The authors showed defect in skeletal muscle repair after multiple round of injury despite the increase SCs fraction. Can you please address if SCs isolated from SlugcKO are precociously activated and consequentially fail to differentiate? Brdu experiments to show expansion due to activation of SCs and analysis of myogenic fate are required. Please quantify the number of myonuclei in regenerating muscle fibers.

After single /double injury test the propensity of Pax7SCs to proliferate/differentiate/senescence.

Figure 4B-C: The author should pull out genes that are involved in muscle SCs self-renewal and maintenance of stem cell pool, if they want to focus on self-renewal defect.

Figure 4H: Please show absolute value of mRNA analysis for p16ink4a level during quiescence and activation in vitro and in vivo in Slug wt versus Slug null. Moreover, a time course analysis of Slug expression during activation in relation to p16 is recommended.

Figure 4I: Please stain for laminin, it is important to know where Pax7+SCs/P16+ cells are

located.

Figure 5 A: Please stain and quantify terminally differentiated myotubes for Myosin Heavy Chain and reserve cells for Pax7. Is there a defect in reserve cell number?

P16ink4a has been used as a marker of senescence, but not all senescent cells are p16ink4a+ (Rodier and Campisi, 2011). Please confirm that increased in senescence is related to increased p16ink4a+ cells.

Please stain reserve cells after 7d subgrowth in vitro with p16 antibody.

Figure 5H: Please quantify Pax7+ senescent cells after 10d post injury.

Their title of figure 5 is misleading, because they did not look at self-renewal, except for the in vitro reserve cells assay. They observed entrance into cellular senescence under proliferative stress. In vivo self-renewal senescence has not been shown, please do that.

Figure 6A-B: After deletion of p16 (again at germline level) is senescence decreased? How many Pax7+ cells are also SAbgal+?

Please quantify self-renewed cells on aged muscle section and SCs contribution to muscle fibers.

SCs repopulation experiments: please specify if your donor SCs are from aged mice and include also young/adult mice as a control.

Figure 7I Quantification of dystrophin+ fibers is needed

Is Slug overexpression impacting cell proliferation and reducing senescence? Provide experimental

Reviewer #1 (Remarks to the Author):

The authors show that slug plays a role for muscle regeneration through regulating p16. They show loss of slug in SCs leads to p16 upregulation and induction of senescence in a p16 dependent manner. Consistently, the loss of slug in SCs results in decreased regenerative capacity during repeated injury. In addition, this is partially rescued by p16 ablation as well as enforced expression of slug. Data suggest a new link between slug and p16/senescence and its key role in muscle regeneration. The experiments are generally well designed.

Supp Fig. 1d shows a reduced myofiber size in whole body slug KO mice, but this doesn't seem to be the case in cKO? The authors might want to comment on this.

Answer: This is an interesting question. It is worth noting that although whole-body Slug KO mice exhibit reduced myofiber size and smaller hindlimb muscle mass than those in wild-type mice; but, when normalized to body weight, none of the relative weights of these muscles were affected by loss of Slug. This is because the whole-body Slug KO mice also exhibit smaller body size, which might be due to involvement of Slug in skeletal stem cell homeostasis and osteogenesis (Tang *et al.*, 2016 Nat Cell Biol 18(9):917). Therefore, we think less muscle mass and smaller myofiber size are only phenotypes in scale with the skeletal bone and body size, rather than reflecting a major role in prenatal skeletal muscle development.

Fig. 3g/h. This is a nice experiment but not quantitative. The figure only shows a single result. They need to show reproducibility of the result.

Answer: Thank the Reviewer for the compliment. This is an adapted competitive repopulation assay that was applied in muscle stem cell study in this manuscript for the first time. We did repeat independently this experiment to have obtained similar result indicating the compromised self-renewing capacity of Slug null SCs (**Supplementary Fig. 6** in this revised manuscript).

Fig. 4c. please provide normalized enrichment scores (NES) and adj-p values.

Answer: Per request by the other reviewer, pathway enrichment analysis displayed in the previous Fig. 4c was revised and presented as **Supplementary Fig. 7** (in this revised manuscript). NES and adj-p values were added correspondingly in each panel.

Fig. 4g. This is not quantitative and they need to show reproducibility. It is also advisable to include negative control regions. Are these cells 'primary' or (somehow) immortalized myoblasts? To conclude that "Slug is a transcriptional repressor for p16lnk4a", they need to provide functional validation using reporter assay with the wt/mutant promoter, otherwise tone down.

Answer: Thanks for the suggestion. We designed another pair of primers targeting the 3' untranslated region with no E-box element as a negative control. The binding affinity of Slug on the putative E-box element was quantified relative to the non-specific binding by ChIP-qPCR assay (**Fig. 4f** in this revised manuscript). For primary myoblasts, they might be somehow immortalized during selection process after CRISPR/Cas9-mediated tagging. As suggested by the reviewer, we performed p16 luciferase reporter assay in freshly isolated SCs to further support the conclusion from the ChIP-qPCR results (**Supplementary Fig. 9** in this revised manuscript).

Fig. 4h: it would be more informative if the authors show the difference between wt resting and wt cultured. Using the ChIP assay, can they see the reduction of slug on the p16 promoter during culture?

Answer: It is a constructive suggestion. We were unable to show reduced binding of Slug on the p16 promoter during culture by ChIP assay because there is a lack of a validated ChIP graded Slug antibody for ChIP assay in freshly isolated SCs. That's why we performed Slug affinity tagging at its C-terminus in SC-derived myoblasts by CRISPR/Cas9 technique to facilitate assessing the occupancy of endogenous Slug at the promoter region of *p16* by ChIP assay. However, we did perform a time-dependent expression analysis of both *Slug* and *p16^{Ink4a}* in *ex-vivo* cultured myoblasts, showing that *Slug* was reduced but *p16^{Ink4a}* was increased along with the culture (**Supplementary Fig. 10** in this revised manuscript). Together with the data from the ChIP-qCPR and p16 luciferase reporter assays, it should be evident that Slug actively represses *p16^{Ink4a}* in SCs.

Fig. 4i,j: *"these results indicated that Slug deficiency leads to an increase in p16Ink4a transcription in SCs, and replicative stress signaling triggered by SC activation and proliferation concurrently increases the stability of p16Ink4a mRNA". This is not supported by the data. Actually, the logic doesn't make sense. How can one hypothesize that mRNA is unstable when it is high? It would make more sense to speculate that p16 protein is unstable in resting SCs?*

Answer: We really appreciate the Reviewer's comments and suggestion. We changed the conclusion into "these results indicated that Slug deficiency leads to an increase in p16Ink4a transcription in SCs, and replicative stress signaling triggered by SC activation and proliferation concurrently increases the stability of p16^{Ink4a} protein." in this revised manuscript. Indeed, it was reported that p16 translation is suppressed by miR-24 (Lal *et al.*, 2008 PLoS One 3(3):e1864), which is highly expressed in quiescent SCs but significantly down-regulated in activated SCs (Sun *et al.*, 2018 Mol Ther Nucleic Acids 11:528). We have added this post-translational regulation mechanism in the Discussion section (in this revised manuscript).

Fig. 5c: *cumulative PD is decreased between p2 and 3, suggesting a substantial number of cells might undergo cell death (or plating efficiency becomes low). The authors need to clarify whether slug ko induces both senescence and cell death thus residual (survival) cells are used for the assays in vitro.*

Answer: Thank the Reviewer for this instructive suggestion. Although decrease of cumulative PD might indicate senescence and cell death or a low plating efficiency, we think that it was more likely ascribed to cellular senescence, due to the following reasons: i) Gene ontology enrichment analysis of biological processes showed that *Slug* deletion derepressed a set of genes related to cellular senescence (**Fig. 4c** in the revised manuscript), but apoptosis-related GO terms were enriched neither in *Slug* null SCs (**Fig. 4c** in the revised manuscript) nor in *ex-cultured* *Slug*-silenced primary myoblasts (**Fig. 4a** in the revised manuscript); ii) New data from SC transplantation experiment in the revised manuscript demonstrated a robust engraftment of *Slug*-deficient SCs after transplantation (**Supplementary Fig. 4e-g** in the revised manuscript). Little or no *Slug* null SCs-derived myofiber should be detected if *Slug* deletion caused cell death in SCs.

"..., muscle repair was significantly improved in Slug-/-p16-/- mice after injuries (Fig. 6a,b)". 'Statistical significance' is not shown.

Answer: We added 'Statistical significance' in **Figure 6b** (in this revised manuscript).

Minor points:

Fig. 4f seems to show tagging of p16, not slug?

Answer: We apologize for this mistake. Flag-tagging was actually introduced in *Slug* gene. The diagram illustrating epitope tagging of endogenous *Slug* in myoblasts by CRISPR/Cas9-mediated

genome editing has been shown in **Supplementary Fig. 8** (in this revised manuscript).

Fig. 7a: which dots represent slug?

Answer: It is indicated with an arrow in each panel (**Fig. 7a**).

--

Reviewer #2 (Remarks to the Author):

In general this is a well written manuscript with significant evidence provided that the EMT transcription factor Slug/Snai2 can control muscle stem cell quiescence/senescence through the repression of p16^{Ink4a}. The SC-specific deletion of a novel conditional Slug allele using Pax-7 Cre mouse line is very convincing concerning the cell-autonomous role of Slug in regulating SC quiescence/senescence. In addition, the rescue of muscle regeneration of Slug deficient mice by breeding onto p16^{Ink4a} deficient background provides very strong genetic evidence of the important of p16 in the Slug null phenotype.

Major concerns:

1) The authors show that there is a difference in Snai2 expression in SC in the muscle between young and old mice and that this correlates with changes in p16 expression. However, no mechanistic insight or discussion is provided concerning exactly how or why Snai2 levels fail off over time in the SC population. Is a similar correlation observed in human muscle samples in young vs old or in control vs diseased muscle biopsies? From a regenerative medicine perspective, how would one potentially modulate Snai1 expression in the adult?

Answer: We appreciate the Reviewer for these constructive comments and suggestions. We did observe an aging-associated reduction of *Slug/Snai2* expression in mouse SCs. Mechanistically, Slug expression is under control of a number of signaling pathways such as fibroblast growth factor (FGF), Wnt, transforming growth factor β (TGF β), Notch, Stem cell factor (SCF), integrins and estrogens *etc.* (Barrallo-Gimeno and Nieto, 2005 Development 132(14):3151). Several of these signaling molecules such as FGF (Chakkalakal *et al.*, 2012 Nature 490(7240):355) and Notch (Conboy *et al.*, 2003 Science 302:1575-1577; Wen *et al.*, 2012 Mol Cell Biol 32(12):2300-2311), which are known as inducers of Slug, were reported to decline with age in mouse SCs. Therefore, we speculate these impaired upstream signaling pathways might account for Slug insufficiency in aged SCs.

Unfortunately, we are unable to provide a direct experimental evidence showing a similar trend of aging-associated decrease of *Slug* in human SCs since our current study is a pilot investigation in mouse, and there lacks referable database comparing gene expression between young and old human muscle stem cells. Nevertheless, we believe our finding in aged mouse SCs may be also conserved in their human counterpart because 1) the potential Slug-binding site (E-box) consensus sequence is also detected in the promoter region of human *p16^{Ink4a}* (**Supplementary Fig. 9c**); 2) Knockdown or over-expressing Slug in primary human myoblasts causes up and down-regulation of *p16^{Ink4a}* expression, respectively (**Supplementary Fig. 9d-g**); 3) active Notch and Notch ligand Delta are declined in old human muscle compared to that of in young's (Carlson *et al.*, 2009 EMBO Mol Med 1:381-391).

Our findings in current study offer a novel therapeutic target for aging-associated degenerative muscle disease. Indeed, there are some small molecules have been reported to induce or suppress *Slug* expression in some types of cells (Barrallo-Gimeno and Nieto, 2005 Development 132(14):3151). We have confirmed that forced expression of activated Notch1 is able to induce Slug expression in cultured myoblasts. Therefore, it would be interesting to conduct *in vivo* studies by intramuscular or intraperitoneal administrating small molecules that induce Slug activators or suppress Slug repressors in aged mice to test the improvement of aging-associated muscle stem cell defects

2) Given the gene changes that occur in Snai2 null SC populations that are involved in cytokinesis, chromosome segregation and microtubule assembly how sure are the authors that asymmetric vs symmetric cell division is not affected and contributing to the regeneration defect given the importance of asymmetric self-renewal in the SC cells of the muscle (see Kuang et al., Cell 2007).

Answer: These are excellent suggestions for the potential mechanistical study on self-renewal defect in Slug-deficient SCs. As revealed by Kuang *et al.* (Cell 2007), Notch signaling plays a critical role in the maintenance of SC self-renewal by asymmetric division. Unfortunately, neither gene ontology analysis nor differential gene expression heatmap results in our Slug null SC data set showed significant changes in components involved in Notch signaling pathways. However, whether Slug directly regulates asymmetric cell-fate determinant Numb segregates is worthy of future investigation. Regarding those significantly changed genes involved in cytokinesis, chromosome segregation and microtubule assembly upon Slug deletion, we did observe a relatively faster activation of Slug-null SCs from G₀ to S stage during first cell division under stimulus. But after a couple rounds of cell proliferation, p16 was significantly increased in Slug-deficient SCs, and caused cellular senescence.

Minor issues:

1) In the discussion the authors mention the possibility of Snai1 partial compensation in Snai2 SC phenotype but do not mention recent paper by Sieiro et al., eLife 2016 whereby it was demonstrated that Snai1 levels can indirectly influence important muscle transcription factors such as Myf5.

Answer: According to the Reviewer's suggestion, we included this paper in the discussion section in our revised manuscript.

2) Figure 4F is confusing and needs to be changed. It is Snai2 locus that has been FLAG tagged at C-terminus and not p16Ink4a. This should be removed as it is depicted in the Sup. Fig 3? In this Figure primers should be indicated that flank E box element. How conserved is this E-box element? Is this E-box element present in the human p16Ink4a promoter (see major concern 1 above).

Answer: We apologize for the mistake. A diagram for epitope tagging of endogenous Slug in myoblasts by CRISPR/Cas9-mediated genome editing has been shown in **Supplementary Fig. 8** (in this revised manuscript). This E-box element is highly conserved, and also present in the human p16^{Ink4a} promoter region (**Supplementary Fig. 9c**).

3) How sure are the authors that the C-terminal tag on SNAI2 that was used for ChIP experiments does not interfere with the normal function of SNAI2? How does this compare to N-terminal tags? Would be good to show that protein half-life and localization is not affected by this Tag.

Answer: FLAG tag is one of the most popular tags and has been widely used to immunoprecipitation and ChIP assays. FLAG tag has never been reported to interfere normal function of target genes. It has been shown that the C-terminal FLAG tag on Snail2/Slug did not affect nuclear localization of Slug in cells and its function in promoting EMT and regulating its target genes (Gastroenterology 2014, 146:1386-1396).

Although we did not compare C-terminal tag with N-terminal tag in our manuscript, we believe that N-terminal tag should work with ChIP assay since it is unlikely for a N-terminal tag to affect the DNA-binding domain of Slug locate in the C-terminal.

We confirmed by western blot that protein half-life was not affected by this tag (**Supplementary Fig. 8d**).

--

Reviewer #3 (Remarks to the Author):

Review of Zhu et al., 2018 Nature Communications manuscript

The manuscript entitled "Transcription factor Slug/Snail2 reinforces self-renewal in aged skeletal muscle stem cell" by Zhu et al., (2018) attempt to demonstrate that the zinc-finger transcription factor Slug is highly expressed in quiescent muscle stem cells, Satellite Cells (SCs), and function as a direct transcriptional repressor of the cell cycle inhibitor, p16ink4a. Then, loss of Slug promotes increases expression of p16ink4a in SCs and accelerates the entry of SCs into a senescent state upon damage-induced stress. Therefore, p16ink4a depletion partially rescues defects of Slug null SCs. The study also, through the body of the results section, infers that decreased level of Slug expression and consequent de-repression of p16ink4a in muscle SCs during aging, intrinsically impact their self-renewal potential after multiple round of regeneration.

It should be noted that the authors of the manuscript under review have previously shown that Slug repress self-renewal of hematopoietic stem cells (HSCs) and is essential for controlling the transition of HSCs from relative quiescence under steady-state condition to rapid proliferation under stress conditions (Sun et al, 2010).

In summary, the manuscript by Zhu et al, (2016) has identified of a novel upstream regulator of p16lnk4a. However numerous technical flaws minimize the authors conclusions

General points:

First, all the experiments are conducted with Slug germline knockout or Pax7-Cre conditional knockout mice, raising the possibility that the phenotype is not confined only to muscle SCs, calling into question the proposed Slug intrinsic role in SCs. Also, the phenotype can derive from post-natal maturation mechanisms that influence adult homeostasis, proliferation, differentiation and self-renewal potential of SCs, during muscle regeneration. So, to this point, it is essential to use Pax7CreER tamoxifen inducible mice models to determine the specific functions of Slug solely in quiescent SCs and their downstream progeny during muscle repair.

Answer: According to the Reviewer's suggestions, we crossed Slug^{fl/fl} mice with Pax7^{creER} mice to obtain Slug^{fl/fl}Pax7^{creER} and Slug^{fl/+}Pax7^{creER} (used as control) mice. *Slug* gene was efficiently deleted in adult MuSC by consecutive intraperitoneal injection of tamoxifen for 5 days (**Supplementary Fig. 3a-c** in this revised manuscript). Similar to what we found in SlugKO (germline knockout) and Slug^{ckO} models (Pax7-Cre conditional knockout), tamoxifen-induced deletion of Slug in adult SCs also caused severe regenerative defect in secondarily but not primarily injured muscles (**Supplementary Fig. 3d-f**).

In addition, we repeated the flow cytometry and transplantation-based quantitative assay for SC self-renewal using primary SCs isolated from Slug^{fl/fl} mice. Slug^{fl/fl} SCs were infected with control retrovirus or retrovirus-expressing Cre recombinase, which deleted Slug efficiently through catalyzing homologous recombination of DNA fragment between loxP sites (**Supplementary Fig. 5a,b**). Control- and Cre-virus-infected cells were then transplanted into each side of pre-injured TA muscles, respectively. Similar to the results using SCs from previous Slug^{ckO} mice, the Cre retrovirus-infected SCs yielded about 5-fold less GFP positive fraction in the total SC population from the recipients compared to that of Control virus-infected SCs (**Supplementary Fig. 5c,d**). More importantly, we confirmed by immunohistochemistry analysis that significantly less Cre retrovirus-infected SCs returned

to native SC niche than Control virus-infected SCs did (**Supplementary Fig. 5e,f**). Together, these data strongly affirmed our previous conclusions based on the Slug^{CKO} mice in this study.

Second, to determine a possible SCs self-renewal potential defects, the author take advantage of transplantation assay and quantify self-renew SCs mainly by FACS analysis of VCAM+/GFP+ SCs from WT or Null mice. As they state in the text, one-month post transplantation some of the donor SCs may be still proliferating, therefore the use of VCAM marker (VCAM marks SCs and their progenitors) cannot be used to determine the fraction of self-renewed SCs. There are two important criteria that SCs must fulfill to be considered 'self-renewed': 1) location: SCs must re-enter the muscle fiber niche and reside under the basal lamina; 2) quiescent state: upon muscle injury SCs activate and acquire proliferation markers (Ki67+/MyoD+). Once they self-renew, they must go back to quiescence (Ki67-/MyoD-). Therefore, the authors need to perform more detailed experiments to determine if Slug impact SCs self-renewal.

Answer: The flow cytometry and transplantation-based quantitative assays for SC self-renewal was performed according to the method described by Arpke and Kyba (2016 Skeletal Muscle Regeneration in the Mouse pp163-179). The similar assay has been also used to compare the cell-autonomous stem cell self-renew capability between young and aged mice (Cosgrove *et al.*, 2014 Nat Med 20:255-264).

According to the paper by Hardy *et al.* (PLoS One 2016 11(1):e0147198), 95% percent of undifferentiated exogenous SCs were quiescent; only a small portion of the donor SCs may be still proliferating even by one-month post transplantation and these cycling SCs could be activated during preparation of mononucleated cells (Velthoven *et al.*, 2017 Cell Reports 21:1994-2004; Machado *et al.*, 2017 Cell Reports 21:1982-1993).

Nevertheless, in response to the reviewer's question regarding if Slug impacts SC self-renewal, we have provided more evidence in the revised manuscript including 1) immunohistochemistry assay demonstrating significantly reduced Slug-null SCs that returned to the native stem cell niche after transplantation in comparison to wild-type control SCs (**Supplementary Fig. 5e,f**); 2) *in vitro* myofiber-associated SC culture assay showing that Slug-deficient SCs have an intrinsic defect in self-renewal (**Fig. 3i-k**).

Third, the authors need to perform a more careful analysis of cell fate during the repair process-while they argue for a senescence mediated control of self-renewal, in vitro experiments suggest that a large number of progenitors undergo senescence-not the cells that return to quiescence. If a large number of progenitors are senescing in vitro-why does this not manifest as a phenotype in vivo? Is this due to the use of germline mouse in senescence experiments, or a decline in the number of myonuclei during regeneration that was not scored by the authors.

Answer: It is notable that *in vivo* regenerating environment is more complicated than *in vitro* culture condition. SCs cultured in serum-rich growth medium mainly undergo proliferation; However, SCs in regenerating muscle go through not only proliferation but also differentiation and self-renewal.

Experimental data obtained from *in vitro* culture model reflected Slug-deficient SCs acquire features of senescence more easily than wildtype SCs under proliferative stress. We showed in the revised manuscript that differentiation of Slug null SCs is normal as indicated by comparative *in vitro* induced differentiation and *in vivo* engraftment after transplantation (**Supplementary Fig. 4c-g**). It is thus reasonable that first regeneration was normal in *Slug*^{-/-} mice. However, when looking into those activated but not differentiated SCs in regenerating muscle, *p16*^{Ink4a} was significantly derepressed after a few rounds of proliferation (**Fig. 4g-i** in this revised manuscript), which was similar to those *ex-vivo* cultured *Slug*^{-/-} myoblasts. These cells had impaired self-renewing capability, and were in senescence (**Fig. 5f-h**) and irreversible quiescence (**Fig. 5i,j**) after first muscle regeneration. Indeed, derepression of *p16*^{Ink4a} has been reported to provoke defective self-renewal in hematopoietic (Smith *et al.*, 2003 Mol Cell 12(2):393-40) and neural stem cells (Molofsky *et al.*, 2005 Genes Dev 19:1432-1437) as well as muscle stem cells (Sousa-Victor *et al.*, 2014 Nature 506:316-321) in *Bmi1*^{-/-} mice.

We also applied tamoxifen-induced adult SC-specific SlugKO mice to confirmed conclusions based on the Slug^{CKO} mice (**Supplementary Fig. 3**) in this revised study.

Fourth, based on their title a reader would expect more figures related to SCs self-renewal decline during aging. Only the final figure attempt to connect Slug, self-renewal and aging. During aging, the number and function of muscle SCs decline. Recently, it has been showed also that maintenance of quiescence in adult mouse life relies on active repression of senescence pathway (Sousa-Victor P et al., 2014), mainly though p16Ink4a. Therefore, does loss of Slug in young cells make them like aged/geriatric SCs?

Answer: This is an excellent question. Many of previous studies from others had already demonstrated that aged SCs are characterized by self-renewal defect (Shefer *et al.*, 2006 Dev Biol 294(1):50-66; Bernet *et al.*, 2014 Nat Med 20(3):265-71; Cosgrove *et al.*, 2014 Nat Med 20(3):255-64) and susceptibility to senescence upon mitogen exposure (Bernet *et al.*, 2014 Nat Med 20(3):265-71; Chakkalakkal *et al.*, 2012 Nature 490(7420):355-60). In addition to evidences including reduced *Slug* and increased *p16^{Ink4a}* expression in aged SCs (Fig. 7a-d), we further demonstrated that *Slug* null SCs resemble aged SCs in both similarly altered metabolic reprogramming and cell cycle regulator signatures (**Supplementary Fig. 12** in this revised manuscript), indicating great similarity in intrinsic propensities of *Slug*^{-/-} SCs and aged SCs.

Detailed points:

Figure 2A: Rather than having SCs compared to DN cells, which are almost unknown population, it will be more informative and relevant to compare population previously characterized such as FAPS, CD31 endothelial cells (where the role of Slug is well characterized) or blood CD45+ cells.

Answer: Following the Reviewer's suggestion, we compared *Slug* expression in endothelial cells (CD31⁺), pan-lymphocytes (CD45⁺), fibro-adipogenic progenitors (CD31⁻CD45⁻Scal1⁺) and satellite cells (CD1⁻CD45⁻Scal1⁻Vcam1⁺) in this revised manuscript (**Fig. 2a**).

Figure 2B: Please quantify slug expression in activated SCs, for example 2-3 days in proliferation conditions. If Slug is increased in proliferating cells, how would this change the authors conclusions?

Answer: According to the Reviewer's suggestion, we quantified *Slug* expression in freshly isolated quiescent SCs (QSC), activated SCs (ASC) upon culture in growth medium for 3 days, as well as the fully differentiated myotubes (MT) (**Fig. 2b** in this revised manuscript). Our data showed that compared to quiescent SCs, *Slug* expression was slightly decreased in activated SCs upon culture for 3 days and fell into a very low level in fully differentiated myotubes. Our result indicated that *Slug* is not increased in proliferating SCs.

Figure 2F: Please quantify Slug+ Pax7+SCs; quantification of Pax7+ SCs need to be more accurate: please stain adult tissue section from Ctrl and Slug cKO mice for Pax7, Laminin and Ki67/MyoD.

Answer: As shown in **Fig. 2f**, all Pax7⁺ SCs in Ctrl mice (*Slug^{fl/+}Pax7^{Cre/+}*) expressed concomitantly *Slug*, while all Pax7⁺ SCs from *Slug^{CKO}* mice (*Slug^{fl/fl}Pax7^{Cre/+}*) were negative for *Slug* staining. However, when assessing efficiency of tamoxifen-induced *Slug* knockout in adult SCs of Ctrl (*Slug^{fl/+}Pax7^{CreER}*) and cKO (*Slug^{fl/fl}Pax7^{CreER}*) mice, quantification of *Slug⁺Pax7⁺* SCs was provided (**Supplementary Fig. 3c** in this revised manuscript).

According to the Reviewer's suggestion, we quantified Pax7⁺ SCs by staining resting adult skeletal muscle sections from Ctrl and Slug^{CKO} mice for Pax7 and MyoD (**Supplementary Fig. 4a,b** in this revised manuscript). We apologize for limiting condition of the fluorescent microscope in our lab, which has only a maximum of three fluorescence channels. Therefore, Laminin was not co-stained in these sections. But based on all other IHC results staining Pax7 and Laminin in current study, all Pax7⁺ SCs were located beneath basal lamina, a classical SC anatomical location (**Fig. 3b** and **Supplementary Fig. 5e** in this revised manuscript). We hope that our answers could satisfy the Reviewer's comments.

Figure 2: The authors showed defect in skeletal muscle repair after multiple round of injury despite the increase SCs fraction. Can you please address if SCs isolated from SlugCKO are precociously activated and consequentially fail to differentiate? Brdu experiments to show expansion due to activation of SCs and analysis of myogenic fate are required. Please quantify the number of myonuclei in regenerating muscle fibers. After single /double injury test the propensity of Pax7SCs to proliferate/differentiate/senescence.

Answer: Similar to wildtype counterparts, SCs in the resting skeletal muscles of Slug^{CKO} mice are quiescent (**Supplemental Fig. 4a** in this revised manuscript). SCs freshly isolated from intact muscles of Slug^{CKO} mice can also robustly differentiate upon induction *in vitro* (**Supplemental Fig. 4c,d** in this revised manuscript) and transplantation *in vivo* (**Supplementary Fig. 4e-g**). We detected a 4-fold increase in the number of SA- β -Gal⁺ cells in transverse sections of Slug^{CKO} TA on day 10 post 1st injury (**Fig. 5g** in this revised manuscript). Pax7 and Ki67 co-immunostaining result demonstrated that these SA- β -Gal⁺ cells were positive for Pax7 but negative for Ki67 staining (**Fig. 5h** in this revised manuscript), suggesting a status of senescence in SCs.

According to the Reviewer's suggestion, we tested the re-activation capacity of Pax7⁺ SCs in Ctrl and Slug^{CKO} mice after first round of muscle regeneration. The results showed that most majority of SCs in Slug^{CKO} mice failed to re-activate as indicated by a markedly lowered percentage of Ki67⁺ SCs on day 2.5 post 2nd BaCl₂ injury comparing to Ctrl mice (**Fig. 5i, j** in this revised manuscript).

Figure 4B-C: The author should pull out genes that are involved in muscle SCs self-renewal and maintenance of stem cell pool, if they want to focus on self-renewal defect.

Answer: Thank the Reviewer for this excellent suggestion. We re-evaluated the gene enrichment analysis results, and demonstrated that there was a switched metabolic reprogramming with relatively higher energy-consuming status in Slug null SC as indicated by enriched glycolysis and oxidative phosphorylation pathways (**Supplementary Fig. 7a,b** in this revised manuscript).

These results indicate that Slug null might disturb balance between self-renewal and differentiation after activation. Indeed, it was previously demonstrated that the mitochondrial-associated metabolism pathway is more silent in Pax7^{Hi} SCs being of higher level of stemness and responsible for self-renewal (Rocheteau *et al.*, 2012 Cell 148, 112-125). In addition, a more recent study using *in vitro* SC culture model showed that enhanced oxidative phosphorylation negatively affects the return to quiescence of activated SCs (Theret *et al.*, 2017 EMBO J 36, 1946-1962).

Figure 4H: Please show absolute value of mRNA analysis for p16^{ink4a} level during quiescence and activation in vitro and in vivo in Slug wt versus Slug null. Moreover, a time course analysis of Slug expression during activation in relation to p16 is recommended.

Answer: We would like to interpret this question as that the reviewer wants to see the fold change of p16^{ink4a} before and after SC activation both *in vitro* and *in vivo*. We still presented the data as relative expression by 2^{- $\Delta\Delta C_t$} method in this revised manuscript, but p16^{ink4a} expression in all other groups were presented in a form of fold change comparing to that of in quiescent SCs of wildtype mice (**Fig. 4g**). Per suggested, we also performed a time course expression analysis of both *Slug* and p16^{ink4a} in *ex-vivo* cultured myoblasts (**Supplementary Fig. 10** in this revised manuscript).

Figure 4I: Please stain for laminin, it is important to know where Pax7+SCs/P16+ cells are located.

Answer: It has been shown that most majority of the Pax7⁺ SCs should have returned to native stem cell niche 30 days after injury (Comprehensive Physiology 2015, 5:1027-1059). Our new data also showed that activated SCs returned to the niche 30 days after transplantation (**Supplementary Fig. 5e**). Our previous Figure 4I (**Fig. 4h** in this revised manuscript) was mainly performed to ask whether p16 protein was elevated in Pax7⁺ SCs after first round of injury.

Figure 5 A: Please stain and quantify terminally differentiated myotubes for Myosin Heavy Chain and reserve cells for Pax7. Is there a defect in reserve cell number? P16ink4a has been used as a marker of senescence, but not all senescent cells are p16ink4a+ (Rodier and Campisi, 2011). Please confirm that increased in senescence is related to increased p16ink4a+ cells. Please stain reserve cells after 7d subgrowth in vitro with p16 antibody.

Answer: Experiments assessing differentiation of Slug-deficient SCs was specifically performed elsewhere in this revised manuscript (**Supplementary Fig. 4c-g**). We did not compare the reserve cell number because we only used the “reserve cell” model to prove that Slug-deficient SCs acquire features of senescence during *in vitro* proliferation. SCs used for this experiment were sorted from Pax7-zsGreen transgenic mice. By day 21 of induced differentiation, equal number of Slug^{+/-}- and Slug^{-/-}- zsGreen⁺ (indicating Pax7⁺) mononucleated cells were sorted again for subculture and SA-β-gal staining experiment.

According to the Reviewer’s suggestion, we demonstrated that the increased senescence in Slug-null reserve cells after culture was related to increased p16^{Ink4a} expression by showing that Slug^{-/-}p16^{-/-} cells exhibited no senescence under the same condition (**Fig. 6f,g** in this revised manuscript). Per suggested, we also stained reserve cells with p16 antibody after 7-day subculture *in vitro*. It turned out that there was significantly higher proportion of p16^{Ink4a+} cells in Slug^{-/-} reserve cell-derived progeny (**Supplementary Fig. 11** in this revised manuscript).

Figure 5H: Please quantify Pax7+ senescent cells after 10d post injury. Their title of figure 5 is misleading, because they did not look at self-renewal, except for the in vitro reserve cells assay. They observed entrance into cellular senescence under proliferative stress. In vivo self-renewal senescence has not been shown, please do that.

Answer: Figure 5h was to demonstrate that the SA-β-Gal⁺ cells detected in **Fig. 5f** were Pax7⁺ SCs. The quantification of these senescent SCs was shown in **Fig. 5g**.

We apologize for the misleading description in title of Figure 5. Quiescent SCs activate and proliferate in both *in vitro* culture and *in vivo* injury conditions. What we had expected to summarize in this title was that lack of Slug facilitates entry of SCs into cellular senescence under proliferative pressure. To this end, we revised the title of **Fig. 5** (in this revised manuscript) as ‘Slug-deficient SCs acquire features of senescence during *in vitro* and *in vivo* proliferation’.

Regarding *in vivo* self-renewal senescence, we performed SA-β-Gal and Pax7 staining on cryosections of TA muscles from Ctrl and Slug^{CKO} mice on day 10 post 1st injury (**Fig. 5f-h**), when most majority of SCs has already self-renewed, and growth of new muscle fibers is very advanced (Dumont *et al.*, 2015 Comp Physiol 5:1027-1059). These results indicated senescence in self-renewed SCs *in vivo*. We further provided *in vivo* experimental evidence showing that self-renewed SCs in Slug^{CKO} mice failed to re-activate upon second injury (**Fig. 5i,j** in this revised manuscript), indicating status of senescence and irreversible quiescence.

Figure 6A-B: After deletion of p16 (again at germline level) is senescence decreased? How many Pax7+ cells are also SAbgal+? Please quantify self-renewed cells on aged muscle section and SCs contribution to muscle fibers. SCs repopulation experiments: please specify if your donor SCs are from aged mice and include also young/adult mice as a control.

Answer: Yes, deletion of p16 is sufficient to significantly decrease senescent *Slug*^{-/-} SCs (less than 1%) under proliferative stress (**Fig. 6f,g** in this revised manuscript).

All the experimental mice used in Fig. 6 were actually young adult mice. Because loss of Slug caused regenerative and self-renewing defects in SCs in young adult mice. We demonstrated that removal of p16 could partially rescue Slug-deficiency caused defects (**Fig. 6** in this revised manuscript).

Figure 7I Quantification of dystrophin+ fibers is needed. Is Slug overexpression impacting cell proliferation and reducing senescence? Provide experimental

Answer: According to the Reviewer's suggestions, we have quantified dystrophin⁺ fibers (**Figure 7j** in this revised manuscript). We also performed gain-of-function studies and showed that Slug overexpression does not impact cell proliferation, but robustly suppressed p16^{Ink4a} expression in long-term cultured SCs (**Figure 7f** in this revised manuscript). As we have added in this revised manuscript, deletion of *p16^{Ink4a}* rescued the cellular senescence phenotype in Slug-null SCs under proliferative stress, , forced expression of Slug should be able to actively suppress cellular senescence. This is also supported by experimental data showing that forced expression of Slug could restore self-renewing capacity of long-term cultured SCs (**Fig. 7g,h** in this revised manuscript).

Reviewers' Comments:

Reviewer #1:

Remarks to the Author:

The authors have adequately addressed the reviewer's questions.

Reviewer #2:

Remarks to the Author:

In general the authors have done a thorough job in addressing my concerns and think that the manuscript is now acceptable for publication.

Reviewer #3:

Remarks to the Author:

This revised manuscript is substantially improved. The authors have addressed all the comments and experiments asked.

I believe the manuscript now it is interesting and suitable for publication in Nature Communication.

However, the authors need to readdress some of the Discussion. The authors discuss Soleimani et al (2012) incorrectly. The authors suggest that Slug (Snai2) does not play a role outside of self-renewal because of the diminishing expression.

Regardless of the expression levels, Soleimani demonstrate that siRNA against Snai2 reduces differentiation. Based on this data it is clear that Snai2 is active in myoblasts. Therefore, as stated, the authors interpretation is incorrect.

Based on Soleimani and many of the in vivo experiments in the current manuscript, it is formally possible that Slug is playing a role outside of self-renewal which could affect muscle regeneration and transplantation potential, by biasing SC proliferation versus differentiation. The authors need to specifically highlight this possibility.

Minor mistake: sentence at line 166 page 6 is confusing.

Reviewer #1 (Remarks to the Author):

The authors have adequately addressed the reviewer's questions.

Reviewer #2 (Remarks to the Author):

In general the authors have done a thorough job in addressing my concerns and think that the manuscript is now acceptable for publication.

Reviewer #3 (Remarks to the Author):

1) This revised manuscript is substantially improved. The authors have addressed all the comments and experiments asked. I believe the manuscript now is interesting and suitable for publication in Nature Communication. However, the authors need to readdress some of the Discussion. The authors discuss Soleimani et al (2012) incorrectly. The authors suggest that Slug (Snai2) does not play a role outside of self-renewal because of the diminishing expression. Regardless of the expression levels, Soleimani demonstrate that siRNA against Snai2 reduces differentiation. Based on this data it is clear that Snai2 is active in myoblasts. Therefore, as stated, the authors interpretation is incorrect. Based on Soleimani and many of the in vivo experiments in the current manuscript, it is formally possible that Slug is playing a role outside of self-renewal which could affect muscle regeneration and transplanted potential, by biasing SC proliferation versus differentiation. The authors need to specifically highlight this possibility.

Answer: We really appreciate all the valuable comments from the Reviewer, which have greatly improved the quality of our current manuscript. We confirmed from the study by Soleimani et al. (2012) that siRNA against Snai2 actually promotes myoblasts differentiation. As these authors described in that paper: "Snai1/2-depleted myoblasts undergo precocious differentiation, as evidenced by significant increase in the expression of myosin heavy chain, a marker for terminal differentiation of muscle cells (Figure 4). Conversely, continuous ectopic expression of Snai1/2 blocked myoblasts' entry into differentiation (Figure S4)".

We agree with the reviewer's suggestion that Slug is likely to play a role outside of self-renewal which could affect muscle regeneration and transplanted potential; because Slug is still expressed in myoblasts, although its expression level is lower when compared with muscle stem cells (satellite cells). Therefore, we have specifically highlighted this possibility in Discussion section in this revised manuscript.

2) Minor mistake: sentence at line 166 page 6 is confusing.

Answer: It has been revised.